



# Seasonal recharge mechanism of the upper shallow groundwater in a long-term wastewater leakage and irrigation region of a river alluvium aquifer

Shiqin Wang[1,2,3], Zhixiong Zhang[1,3], Shoucai Wei[4], Wenbo Zheng[1], Binbin Liu[1], Matthias Sprenger[5], Yanjun Shen[1], Yizhang Zhang[6]

[1]Key Laboratory of Agricultural Water Resources, Center for Agricultural Resources Research, Institute of Genetics and Developmental Biology, Chinese Academy of Sciences, Shijiazhuang 050021, China

[2]Xiongan Institute of Innovation, Chinese Academy of Science, China

[3]University of Chinese Academy of Sciences, 19A Yuquan Road, Beijing 100049, China

[4]Binzhou University, Shandong Provincial Key Laboratory of Eco-Environmental Science for Yellow River Delta, Binzhou 256603, China

[5]Earth and Environmental Sciences, Lawrence Berkeley National Laboratory, 1 Cyclotron Road, Berkeley CA 94720, USA

[6]Chinese Research Academy of Environmental Sciences, 8 Dayangfang, Anwai Beiyuan, Beijing 100012, China

*Correspondence to:* Shiqin Wang (sqwang@sjziam.ac.cn); Yizhang Zhang (zhangyz@craes.org.cn)

**Highlights:**
- ➤ Groundwater recharge and contamination show obvious seasonal variation
- ➤ Two types of hysteresis loops in $\delta^2H$ and $\delta^{18}O$ plots of groundwater were found
- ➤ Mixing of fast flow are major processes of groundwater recharge
- ➤ Residual pollutants in soil account for variation of groundwater quality

**Abstract.** Understanding the mechanisms that control seasonal groundwater recharge at local and intermediate scales is critical for understanding contaminant transport. Preferential flow accompanied with intensive and seasonal recharge allows contaminants to migrate rapidly through the unsaturated zones to underlying aquifers. In this study, we investigated the recharge mechanism from multiple water sources to the upper shallow groundwater of river alluvium aquifers at the Tanghe Wastewater Reservoir (TWR) with a length of 17.5 km in the Xiong'an New Area, North China Plain. To do so, we sampled 30 m deep soil porewater profiles and groundwater boreholes perpendicular to the TWR. We traced the recharge processes and related pollutants transport pathways based on stable water isotopes ($^2H$ and $^{18}O$). The stable isotopes in porewater of the soil profiles revealed vertical recharge rates ranging from 63 to 109 cm/year for the layered unsaturated zone with silt and silty clay loams. However, fast flow (i.e., preferential flow or lateral flow) occurred in sand layers of river alluvium aquifers with the mixing of slow translatory flow with multiple recharge sources (i.e., precipitation, irrigation water, and wastewater remaining in porewater). The distribution of $\delta^2H$ and $\delta^{18}O$ relationship of the upper shallow groundwater showed two hysteresis loops which reflected the impact of fast flow with seasonal variation and translatory flow with legacy water in soil: 1) groundwater in regions affected by the TWR wastewater leakage shows a narrow loop and a nearly straight line with end-members of precipitation, which recharged to groundwater by fast flow, and evaporated porewater crossing with the TWR evaporation line; and 2) groundwater in irrigated farmlands with low and high irrigation amounts and strong evaporation shows

a concentrated loop overlapping with shallow porewater, suggesting the impact of porewater persists in shallow soil on groundwater. The recharge type of fast flow determines seasonal variation of groundwater $SO_4^{2-}$ and $NO_3^-$. The $SO_4^{2-}$ concentrations in groundwater were diluted after recharge of fast flow and then increased due to the contribution of slow flow in porewater. However, $NO_3^-$ increased as the fast flow carried pollutants from shallow soils to the aquifer. Measures are

needed to prevent contamination caused by preferential flow in river alluvium aquifers at local scales which can extend to regional scales.

**Keywords:** recharge processes, seasonal hysteresis, stable isotopes, wastewater leakage, river alluvium aquifer

## 1 Introduction

Groundwater is the primary water supply in many regions that have rapidly expanding water requirements including urban,

industrial, and agricultural production. Groundwater is particularly important in alluvium aquifers, where groundwater renewal rates are generally high (Sun et al., 2021; Ma et al., 2019) and thus, functioning as potable water sources in arid, semi-arid, and semi-humid areas. However, these aquifers are also susceptible to depletion and contamination, with recharge rate and dominant flow processes determining their level of vulnerability (De Vries and Simmers, 2002). Groundwater recharge is influenced by complex ecohydrology processes that are controlled by geology, meteorology, morphology, and vegetation.

Identifying recharge sources and mechanisms is important to develop strategies to prevent groundwater pollution (Jasechko, 2019).

Precipitation or other recharge sources show seasonal characteristics in these areas, which result in seasonal variation of recharge rates, fluxes, and thus transport of potential contaminants (Tweed et al., 2020). Many studies showed that the extent of attenuation and lapse effects on groundwater recharge rates differ by season (Ajami et al., 2012; Jasechko, 2019; Mackay

et al., 2020). Understanding recharge rates at seasonal timescales can help improving forecasts of long-term changes of annual recharge rates under changing climate conditions (Niraula et al., 2017; Raihan et al., 2022; Yang et al., 2021). The seasonal stable isotopic ($^2$H and $^{18}$O) signal of natural recharge disappears in groundwater when the water was moved through the low-conductivity vadose zone to the aquifer leading to intense mixing (i.e., dispersion) along the flow path. However, when water travels through a thin and coarse size stratum, or preferential flow occurs in the unsaturated soil, the natural recharge maintains

the isotopic signals characteristic of seasonal variation, which can then also be detected in the groundwater.

Non-natural recharge from leaky infrastructure, storm water routing, or dry season irrigation (Grande et al., 2020) also contribute to groundwater recharge at local and intermediate scales. While the recharge volumes may not be important for water resource assessment, it is critical for tracking contaminant transport in that concentrated recharge and preferential flow allow contaminants to rapidly migrate through the unsaturated zone to underlying aquifers. Industrial or domestic effluent is

often recharged to aquifers through wastewater treatment plants (Mccance et al., 2020), sewage reservoirs, polluted rivers



(Beckers et al., 2020), sewage-irrigated farmland (Wu et al., 2015), or broken drainage (Ishii et al., 2021). Long-term sewage infiltration can cause a degradation of the groundwater quality. Particularly, unlined wastewater reservoirs or ponds can lead to the deterioration of regional scale groundwater quality (Gal et al., 2009; Wang et al., 2014).

Stable isotopes of water are efficient tracers for describing and quantifying groundwater recharge rates, water flow pathways, response times, and mechanisms under natural and non-natural conditions (McGuire et al. 2002, Healy and Scanlon, 2010, Tekleab et al. 2014, Beddows et al. 2016, Koeniger et al. 2016; Tipple et al. 2017, Vystavna et al. 2019, Solder and Beisner 2020). The isotopic composition of precipitation varies seasonally, reflecting seasonal shifts in moisture sources, air mass trajectories, and cloud processes (Craig 1961, Dansgaard 1964, Rozanski et al. 1993). Coupling precipitation time series to vertical stable isotope porewater profiles in the unsaturated zone (Lee et al., 2007; Garvelmann et al., 2012; Sprenger et al., 2016) and stable isotopes in groundwater (Ma et al., 2017; Wright and Novakowski, 2019) have been used to investigate seasonality and residence time of soil water and groundwater. The processes involved in groundwater recharge such as mixing and preferential flow have also been thoroughly researched (Lee et al., 2007; Ma et al., 2017).

Evaporation causes enrichment of heavy isotopes of water, which differs from meteoric waters (e.g., infiltrating snowmelt or rainfall). The signals of evaporation distillation of stable isotopes in water can be used to tracing the processes of groundwater recharge. Benettin et al. (2018) reported that the reference of the original composition of a water source according to the evaporation line should be valid if the evaporated samples all originate from a single source water. However, for different seasonally varying isotopic sources, numerical experiments based on established isotope fractionation theory have suggested that residual water samples show a hysteresis loop (Benettin et al., 2018). It has also been pointed out that even where such hysteresis loops exist in nature, they may be difficult to detect due to measurement uncertainties and environmental noise. Therefore, it is difficult to quantify the seasonal variation of evaporation and corresponding recharge processes under varying sources conditions.

Previous research has identified seasonal recharge (Ma et al., 2017) and seasonal contamination (Vystavna et al., 2019) in alluvium aquifers located in temperate climates using a combination of stable isotopes and hydro-chemical methods. In some alluvium aquifers, pumping irrigation water from groundwater sources causes groundwater levels to decline and irrigation-return flow then can experience strong seasonal evaporation effects which alters the water cycle. Intense wastewater irrigation and leakage accelerates infiltration and recharge of water and transport of solutes to the groundwater (Gal et al. 2009). This is known to occur in river alluviums where aquifer permeability is higher than other sediment alluviums. Under the impact of multiple water sources (precipitation, wastewater leakage, and irrigation), little is known about the seasonal impacts of these multiple water sources on evaporation of soil water and groundwater, and therefore on groundwater recharge.

In this study, a linear wastewater reservoir in the Xiong'an New Area, in the Baiyangdian Lake watershed of the North China Plain (NCP), was selected as a case study. The reservoir is unlined and was built along a river channel for storing industrial sewage and wastewater from 1977 to 2015. Over-pumping of groundwater for irrigation has led to groundwater decline in the

alluvium aquifer of the NCP (Wang et al., 2008; Zhang, 2009). Therefore, the wastewater was stored in the reservoir and lost

as evaporation, leakage, and agricultural irrigation. The impact of the wastewater reservoir on shallow groundwater has been

studied by using an evaporation and recharge model based on isotope fractionation theory (Wang et al., 2014). However,

the recharge processes with varying and multiple recharge sources was still unclear. Due to evaporation, the isotopes in the

wastewater have a strongly enriched signal which can be used to trace which processes are occurring during groundwater

recharge.

This research applied a multi-tracer method to study the seasonal mechanisms of groundwater recharge as affected by a long-

term sewage leaked under a complex system of natural and non-natural recharge sources (sewage irrigation and groundwater

irrigation). The aims of this paper are to understand the mechanisms that control seasonal groundwater recharge in alluvial

aquifers affected multiple recharge sources.

## 2 Materials and methods

### 2.1 Study site

Our study site was the Tanghe Wastewater Reservoir (TWR) in the Xiong'an New Area of the Baiyangdian Lake watershed

(Fig. 1a). The reservoir is unlined and was built along a river channel (Tanghe river) for storing untreated industrial sewage

with high concentrations of sulphate from Baoding City for almost 38 years (from 1977 to 2015). The north embankment of

the TWR was same with the Tanghe river embankment, while the south embankment was constructed in the Tanghe river

channel. The TWR has a total length of 17.5 km, width of about 100 m water surface, and a storage capacity of $7.2 \times 10^6$ m$^3$.

It was built along the northern part of the dried Tanghe River Channel (Fig. 1c) that flowed into the Baiyangdian Lake and is

separated by a sluice gate. The stratigraphy of the study area consists of Quaternary alluvial deposits with alternating deposits

of silt, silty clay and sand (Zhang, 2009). The site is characterized by a continental monsoon climate with a strong seasonality.

The weather is cold in the winter and warm in summer with an annual mean temperature of 13 °C. The annual mean

precipitation is 504 mm/year, of which more than 75% occurs between July and September (rainy season) as heavy rainfall

events (Fig. 1b). The potential evapotranspiration is 1208 mm/year and the evaporation from March to June is higher than

during other months. Winter wheat and summer maize are the major land uses around the TWR (Fig. 1 c and d). The wastewater

from the TWR was mainly used for agricultural irrigation during the past several decades. In recent years, shallow groundwater

(80-120 m depth) was used for irrigation. At a regional scale, over-pumping of groundwater has led to the drying of natural

rivers and lakes, and the decline of groundwater levels. However, artificial water recharge to the Baiyangdian lake increased

the water table around the lake, the groundwater level in the area around TWR shows a high level in the east and a low level

in the west (Yuan et al., 2017; Yuan et al., 2020). In regional scale, groundwater flows from northeast to southwest with a

hydraulic gradient of 0.5‰-1.3‰ at the middle of TWR in Fig. 1c (by kriging interpolation with the groundwater level data



measured in November 2018). Along with the irrigation return flow of sewage and shallow groundwater, infiltration of wastewater is another major source for groundwater along the TWR. It has been estimated that 76% of the sewage recharges

into an aquifer by leakage and irrigation and the influencing range was about 3 km around the TWR and 150 m depth of the aquifer (Wang et al., 2014).

After the TWR started storing sewage in 1977, the impacts of anthropogenic activities on the TWR varied greatly (Fig. 2). The amount of sewage entering the reservoir was reduced and stopped in the middle part of the channel in 2010. The sewage inflow was totally intercepted in 2015. Due to the development of the Xiong'an new area in 2017, the sewage that remained in the

reservoir was removed in 2018.

Our study area was located in the middle of the TWR, which has a large industrial factory near the observation profiles north side (Fig. 1d). Additionally, there were many small factories in the north of the TWR according to the field investigation (not shown in Fig.1). Crops are cotton and wheat/maize in the north and south, respectively. In April, 2018, cotton was converted into woods, and grass was planted in the TWR channel (Fig.S1).


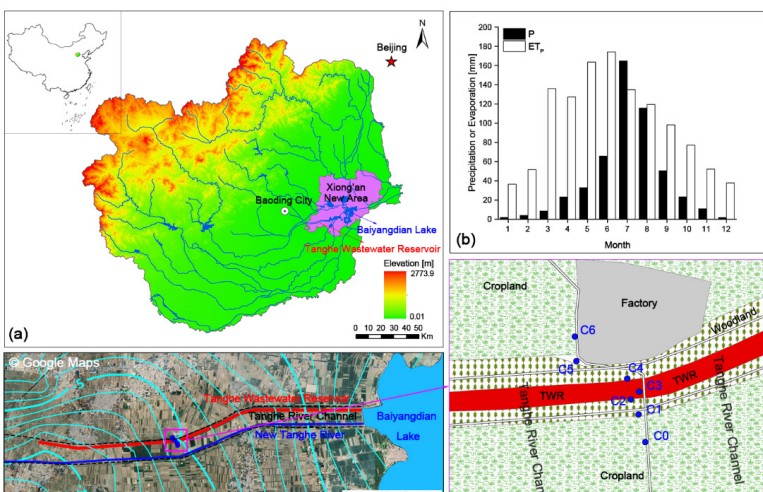

**Figure 1.** Geographical location of the Tanghe Wastewater Reservoir (TWR) and soil sampling sites. **(a)** Position of the TWR in the Baiyangdian Lake watershed. **(b)** The mean monthly precipitation (P) and potential evapotranspiration (ET$_P$) over the study period (1977–2019). **(c)** Position of the study transect in the middle of the TWR and the flow map of the unconfine aquifer around TWR in November

2018. **(d)** Sampling sites along the transect.



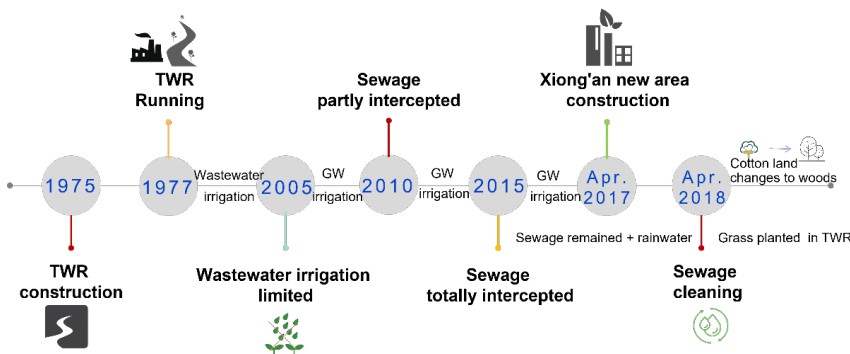

**Figure 2.** Impact of anthropogenic activities on the TWR over the past four decades.

## 2.2 Field work and sampling

This study focused on a 480 m long transect perpendicular to the TWR along which we took porewater profiles and
groundwater samples. The transect was located 8 km upstream from the west outlet of the TWR (Fig.1 c and d). There were
seven soil profiles with a depth of 30 m, drilled with a powered soil core drill along the transect during 8th to 19th June 2017.
From south to north, the soil profiles are referred to as C0, C1, C3, C5, and C6 (Figure 1c). The C3 profile was located in the
middle of the TWR channel, where sewage remained during the sampling period. C1 and C5 were located outside of the north
and south embankments of the TWR, respectively. Profiles C0 and C6 were about 150 m distance from the reservoir
embankments and were located in agricultural fields growing wheat/maize and cotton, respectively. These two profiles
represent the soil profile affected by both sewage seepage and irrigation.

Soil samples of five profiles (excluding C2 and C4) were collected at 10, 20, 30, 50, 70 and 100 cm to a depth of 1 m, at 0.5
m intervals from 1 to 10 m depth, and at 1.0 m intervals from 10 m to 30 m depth, with additional soil samples taken at depths
where the soil texture changed. Soil samples collected using aluminum boxes were primarily used to determine the gravimetric
soil moisture content and soil texture. Soil samples were collected and sealed in small brown glass vials and stored in a -20°C
refrigerator until the analysis of stable isotopic compositions of porewater. Soil samples were packed into plastic Ziploc bags
for analysis of major water chemical ions. In addition, undisturbed 0.1 m soil cores were collected using a ring sampler and
used to determine the physical parameters of the soil, including bulk density, porosity, saturated hydraulic conductivity, and
bulk weight.

After collecting the soil samples, polyvinyl chloride (PVC) tubes with 30 m depth were installed into the borehole to construct
the groundwater monitoring wells. The screen depths were at 15-25 m depth. Groundwater samples of the monitoring wells
(C0, C1, C3, C5, and C6) were collected monthly from October 2017 to June 2019. We sampled wastewater in June 2017 and
irrigation water from shallow groundwater from 80-120m depth and deep groundwater from 120-300 m in September 2009
(Wang et al., 2014). All water samples were collected in polyethylene plastic bottles (50 mL) and brought to the laboratory



for analysis of major cations, anions (i.e., $Cl^-$, $NO_3^-$, and $SO_4^{2-}$) and stable isotopes. The groundwater level was determined during each sampling using a water level measuring ruler. The pH, electrical conductivity (EC), oxidation–reduction potential (ORP), and temperature were measured in situ using a portable meter (WM-22EP, DKK, TOA Corporation of Japan).

### 2.3 Chemical and isotope analysis

We derived the gravimetric soil moisture content via oven drying (105°C, 12 h) and converted this to soil moisture content

based on the measured bulk weight. Soil bulk density was estimated by relating the oven-dried soil weight to the soil sample core volume.

Soil samples for analysis of major ions ($Na^+$, $K^+$, $Ca^{2+}$, $Mg^{2+}$, $Cl^-$, $SO_4^{2-}$, and $NO_3^-$) were dried and ground, and chemicals in solution were determined by extracting soil using a mass ratio of soil and ultrapure water of 1: 5. All water samples were filtered through 0.45 μm and 0.2 μm filters before analysis. Major ions were measured using ion chromatography (ICS-2100,

Dionex, U.S.A.).

Soil porewater was extracted by the automatic water extraction system (LI-2100, LICA, China) for 3 hours, separated by evaporation at 125°C and collected by condensation at -94°C with a system pressure of 1000pa. We analyzed the extracted porewater for their oxygen and hydrogen isotopes at the Key Laboratory of Agricultural Water Resources, Chinese Academy of Science via a laser absorption water-vapor isotope analyzer (Picarro-i2120, CA, U.S.A.). The stable isotope ratios are

expressed in delta (δ) units and a per mil (‰) notation relative to the Vienna standard mean ocean water (VSMOW). The reported analytical errors for oxygen and hydrogen isotopes ratio are ±0.1‰ for $\delta^{18}O$ and ±1‰ for $\delta^2H$, respectively.

### 2.4 Data analysis

### 2.4.1 Estimation of groundwater residence time and response time

We estimated the mean groundwater residence time based on the sine wave approach fitting the seasonal signal of water

isotopes in rainwater and groundwater (Mcguire et al., 2002; Tekleab et al., 2014). The periodic regression analysis was performed by the trigonometric function. The predicted water isotopes were defined as follows (Beddows et al., 2016):

$$\delta = \delta_0 + \delta_A \sin \frac{(t-t_0)2\pi}{\tau} \qquad \text{Eq. (1)}$$

where δ is either $\delta^2H$ or $\delta^{18}O$, $\delta_0$ is the sine offset on the δ axis, $\delta_A$ is the amplitude of the sine function, t = date, $t_0$ = date where the sine phase equals zero, and $\tau$ is the period of the sine function.

Groundwater response time refers to the time required for a groundwater system to change from some initial condition to a new state, and is commonly used to indicate the lag time for groundwater quantity or quality to respond to surface recharge (Jazaei et al., 2014). We inferred the response time ($\Delta t$) (Beddows et al., 2016) and residence time (T) (Tekleab et al., 2014) from the fitted sine wave in the input and output signals as

$$\Delta t = t_{02} - t_{01} + n\tau \qquad \text{Eq. (2)}$$




$$T = \frac{\tau}{2\pi} \sqrt{(\frac{\delta_{A1}}{\delta_{A2}})^2 - 1} \qquad \text{Eq. (3)}$$

where $t_{01}$ and $t_{02}$ are the dates where the sine phase equals zero for precipitation and groundwater, respectively, n is the number of periods of the actual response time of the precipitation and groundwater signals, $\delta_{A1}$ and $\delta_{A2}$ are the amplitudes of the sine function for precipitation and groundwater, respectively.

### 2.4.2 End member mixing analysis

We applied end-member mixing analysis (EMMA) to the isotopic and hydro-chemical compositions of the groundwater samples to identify and quantify the fraction of different recharge sources.

This technique assumes that: (1) groundwater is a mixture of sources that have relatively constant composition; (2) the mixing process is linear and completely dependent on hydrodynamic mixing; (3) the substances used as tracers are conservative; and (4) the sources have distinct concentrations (Barthold et al., 2011). The equations are as follows:

$$f_A + f_B + f_C = 1 \qquad \text{Eq. (4)}$$
$$\delta_A f_A + \delta_B f_B + \delta_C f_C = \delta_s \qquad \text{Eq. (5)}$$
$$C_A f_A + C_B f_B + C_C f_C = C_s \qquad \text{Eq. (6)}$$

where δ represents the isotopic ratio, C is concentration of Cl⁻, subscript S represents the groundwater sample, subscripts A, B and C are the three considered end-members, and f represents the fraction of an end-member.

## 3 Results

### 3.1 Constructions of the section profile

Drilling of geological boreholes revealed multiple layers of silt, silty clay, and sand of varying thicknesses (Fig. 3). The
elevation of the transect ranged from 8 to 10 m above sea level. There were two sand aquifers above 30 m depth. The depth of the first aquifer ranged from 8 to 12 m and was thickest beneath the TWR channel (Fig. 3). The depth of the second aquifer ranged from 22 to 28 m, with a thickness of about 2 m. The aquifer depth on the north site (C6) was shallower than the south sites (C1 and C0). The silt layer on the northern site was thicker than on the southern site. A silt layer under the TWR channel constituted a weakly permeable aquifer that connected the sewage to the TWR. A field survey found that black/brown
sediments extended along this silt layer from north and south, suggesting a possible lateral pathway of sewage transport due to the Fe or Mn contained sewage. Layers of silt and silt loam were distributed throughout the soil profile. The monthly variations in water table levels for all monitoring wells were calculated according to the elevation of each well and monitored groundwater levels (Fig. S2). At the beginning of the monitoring period, groundwater levels along the transect decreased from the north sites to the south sites (C6>C5>C3>C4>C2>C1>C0), indicating a groundwater flow from north to south. However,
the direction of the groundwater flow in the north side of the TWR reversed after April 2018. The high water level of the lake in the west resulted in the lateral recharge which changed the groundwater flow.

To identify the stable isotopes characteristics, the depth profiles were divided into three groups (Fig.3): (I) The top silt layer (above the first silty clay layer, where there was little evidence of the sewage leakage); (II) From the top of first silty clay layer to the top of first sand layer, where sewage might penetrate the silt and silty clay reaching the first aquifer; and (III) Depth

from the top of first sand aquifer to the bottom of the second aquifer.

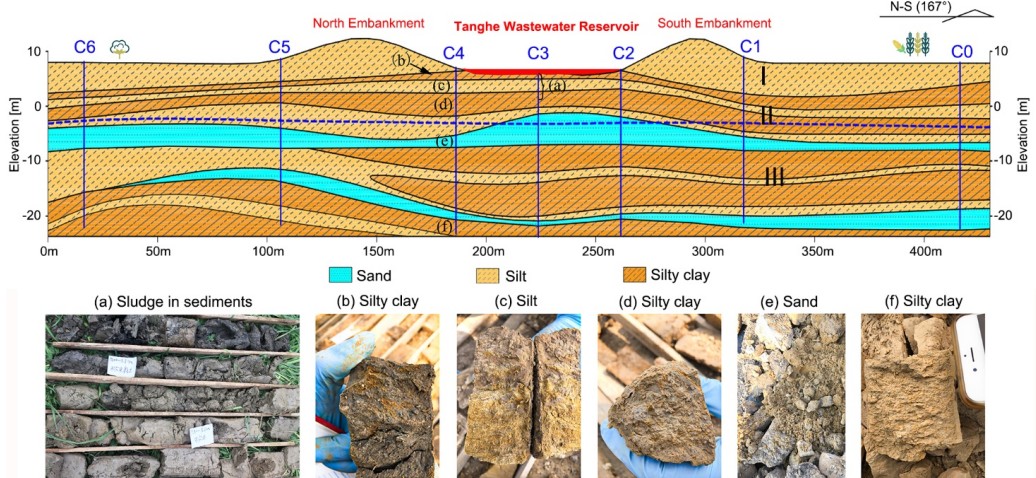

**Figure 3.** Profile along the section in the middle part of the TWR. The dashed blue line is the groundwater level. The blue vertical lines and naming C0 – C6 represent the location and depths of the boreholes and depth profiles, number I, II, III represent the three groups defined in

the text, and the photos (a) to (f) represent extracts from the boreholes with their locations identified with the letters in the vertical profile. Soil profiles C0-C6 are projected in the direction perpendicular to TWR.

### 3.2 Water chemicals and stable isotopes in wastewater

The stable isotopes of water and major water ions of sewage in the TWR at different times are presented in Fig. 4. The stable isotopes in sewage in the TWR show $\delta^2H$ and $\delta^{18}O$ compositions of -50.9 and -6.9‰ in Sep. 2008, respectively, and -48.3 and

-5.6‰ in June 2009. As there is no inflow sewage after 2015, the sewage progressively evaporated, whereby the $\delta^2H$ and $\delta^{18}O$ compositions enriched to -34.8 and -2.6‰, respectively, by June 2017. The high concentrations of $Na^+$ and $SO_4^{2-}$ were indicative of sewage in the TWR. The $Na^+$ and $SO_4^{2-}$ concentrations increased from September 2008 to June 2011 (i.e., $SO_4^{2-}$ concentrations increased from 1528.3 to 3171.0 mg/L). After the TWR interception in 2015, the concentration of $Na^+$ and $SO_4^{2-}$ decreased to 280.9 and 369.1 mg/L, respectively, in June 2017.

All sewage isotopes plotted closely around the TWR evaporation line ($\delta^2H = 4.93 \times \delta^{18}O - 20.99$, $R^2 = 0.99$), which was estimated from the isotopes of wastewater collected by previous work (Wang et al., 2014). Wang et al. (2014) reported that the sewage stored in the TWR was enriched in heavy stable isotopes due to evaporation. As $Na^+$ is easily absorbed in clay or silt clay, the indicative ions of $SO_4^{2-}$ companied with the enriched isotopes of sewage will be applied as tracers to investigate the



impact of the sewage on porewater and groundwater.

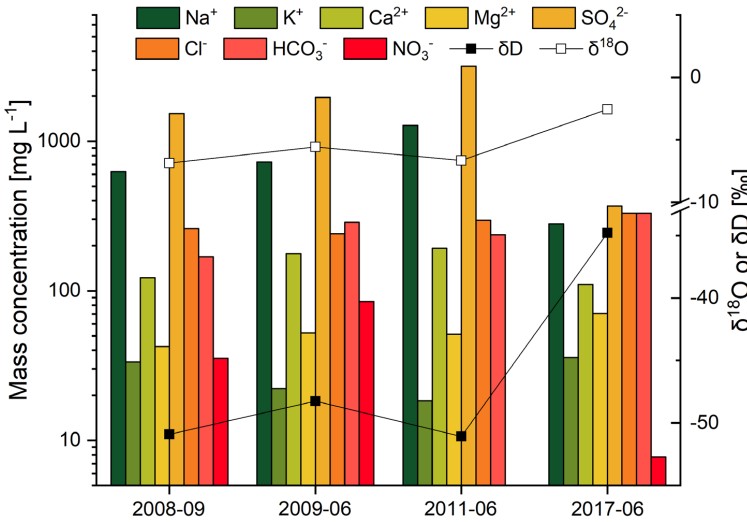

**Figure 4.** Variation of stable water isotopes and major water chemicals in sewage of the TWR.

### 3.3 Stable isotopes in porewater

The variations of $\delta^2H$ and $\delta^{18}O$ in porewater throughout the soil profiles of C0, C1, C3, C5 and C6 are presented in Fig. 5. There were large variations in the pore water stable isotope signatures of the shallow 0-600 cm and deep 2250-3000 cm depths. The large variations in stable isotopes at shallow depths between the sites reflected the different surface condition at different sites. Variation in porewater isotope decreased with depth at all sites, while the most depleted isotope values were found in the silt clay layer at about 2250-3000 cm depths at C6 and to lesser extend at C1 and C5.

The stable isotopes of different sites show gradual enrichment (C0, C1, C6) or depletion (C3) trend with depths due to the infiltration of different water sources. The stable isotopes in the shallow soil of the C3 profile (0-300 cm, excluding 60 cm) were most enriched and least variable with $\delta^2H$ ranging from -49.1 to -47.2‰ and $\delta^{18}O$ ranging from -5.2 to -4.1‰. Then the stable isotopes showed a depleting trend from 300 to 650 cm with peak values of -61.9 and -7.4‰ for $\delta^2H$ and $\delta^{18}O$, respectively. The isotopic values then decreased at a relatively stable rate below 650 cm depth (Group III). These variations are consistent with variations of the depleting isotopes in sewage in the TWR backward from 2017 (Fig. 4). Differing from the C3 profiles, the stable isotopes in C0 and C6 increased from the surface (Group I) with increasing depths (Group III), except for the silty clay depths deeper than 22.5 m. This is consistent with the increasing variation in isotopes from irrigation water with shallow groundwater at present (mean $\delta^2H$: -62.5‰; mean $\delta^{18}O$: -7.9‰) and with wastewater (mean $\delta^2H$: -46.0‰; mean $\delta^{18}O$: -5.4‰) historically. The ranges in isotopes in C6 (from -88.3 to -43.9‰ and from -10.6 to -2.25‰ for $\delta^2H$ and $\delta^{18}O$, respectively) were larger than in C0 (from -70.4 to -45.7‰ and from -8.8 to -2.9‰ for $\delta^2H$ and $\delta^{18}O$, respectively). The mean isotopic values in Group I and III in C6 were greater than in C0. This discrepancy can be attributed to the different infiltration mechanisms




for irrigation water in the cotton and wheat/maize, the evaporation in the cotton land with smaller irrigation (100 mm/a (Min

et al., 2018)) caused the isotopes in soil water more enrichment that those in the wheat/maize cropping system with larger

irrigation (320-360 mm/a (Sun et al., 2006; Zhang et al., 2018)). The C1 and C5 profiles were located outside of the TWR

embankment. The stable isotope distributions in C1 and C5 had greater enrichment with greater depth, which was also observed

in C6 and C0, excluding the isotopes in the silt clay layers at deep depths. This is consistent with the impact of precipitation

that has a more depleted isotopic signature than sewage.

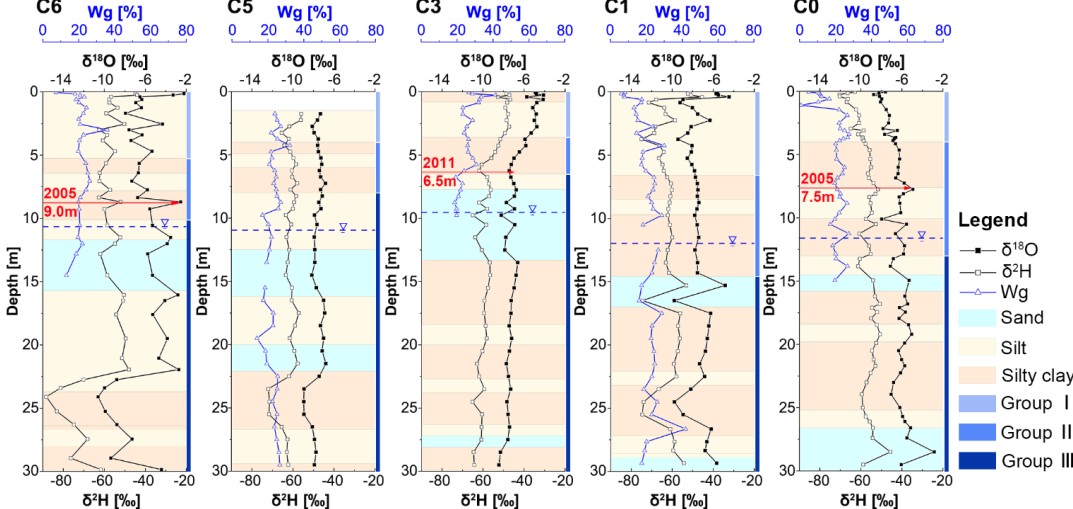

**Figure 5.** Vertical distribution of stable isotopes and gravimetric water content (Wg) in porewater for profiles C0-C6.

There were similarities in soil profile porewater $\delta^2$H and $\delta^{18}$O trends in C1, C3, and C5, and in C0 and C6 (Fig. 6). All profiles

showed isotopic values that were distributed under the evaporation line of the TWR ($\delta^2$H = 4.93 × $\delta^{18}$O − 20.99, $R^2$ = 0.99)

(Wang et al., 2014). The slope of the fitting line for isotopes in C3 and C1 were similar to that of TWR, with slopes of 4.87

and 4.65, respectively (C3: $\delta^2$H = 4.65 × $\delta^{18}$O − 25.68, $R^2$=0.95; C1: $\delta^2$H = 4.87× $\delta^{18}$O − 25.54, $R^2$=0.95), suggesting the

impact of partly evaporated wastewater on porewater. There were few points in the aquifers where C1 showed extreme

deviation from the TWR evaporation line (labeled in Fig. 6a). The samples taken from C5 were clustering in the similar location

in the dual isotope space as Group III of C3, and Groups II and III of C1 (purple solid line in Fig. 6a). This excluded several

samples from the deep silt clay layers (purple dashed line in Fig. 6a). Samples from C0 and C6 were not distributed close to

the TWR evaporation line (Fig. 6b). Samples from C6 showed greater deviation from the evaporation line than C0 (C6: $\delta^2$H =

2.60 × $\delta^{18}$O − 41.72, $R^2$=0.79; C0: $\delta^2$H = 4.13× $\delta^{18}$O − 30.90, $R^2$=0.83). All of these samples deviated far from the LMWL

and the TWR evaporation line suggesting the influence of strong evaporation prior and/or during the infiltration of wastewater





irrigation at farmlands. We further found that the porewater samples from the sand aquifers at each site were most isotopically

enriched and samples from deep silt clay layers were most depleted in heavy isotopes.

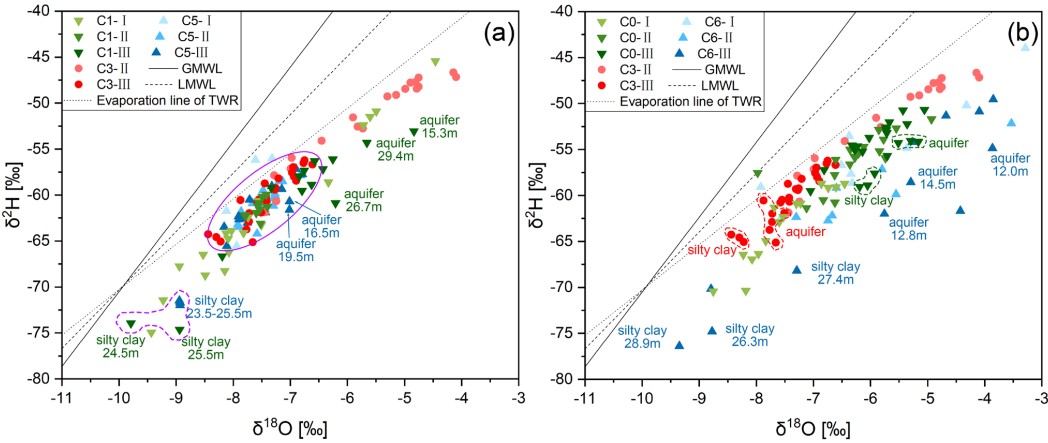

**Figure 6.** Relationship of $\delta^2H$ and $\delta^{18}O$ in porewater throughout the soil profiles. **(a)** Soil profiles C1 and C5 compared with C3; **(b)** Soil

profiles C0 and C6 compared with C3. The number of I, II, III in legends were groups of different layers as defined in Fig.3. The global

meteoric water line (GMWL) is from (Craig, 1961). The local meteoric water line (LMWL) was obtained from the precipitation data for

Shijiazhuang city which is near the study area. The TWR evaporation line is: $\delta^2H = 4.93 \times \delta^{18}O - 20.99$, $R^2 = 0.99$ (Wang et al., 2014).

**3.4 Stable isotopes in groundwater**

The stable isotopes in precipitation and groundwater showed seasonal variation from October 2017 to July 2019 (Fig. 7). The

isotopes in the rainfall in the North China Plain were isotopically depleted in the summer and the winter due to the differing

effects of air moisture sources (Yamanaka et al., 2004). The input signal of isotopes displays the two peaks with high isotope

values during the irrigation season (spring and autumn) (Fig. 7a). Shallow groundwater (at 80-120 m depth) used for irrigation

also had a relatively depleted isotopic signature (mean $\delta^2H$: -62.5‰; mean $\delta^{18}O$: -7.9‰). Corresponding to this, the peaks of

depleted isotopes in groundwater occurred in April during spring irrigation and in October after the rainy season (Fig. 7b). The

seasonal variation of recharge sources had an impact on groundwater.

When comparing the groundwater of the five sites, groundwater at C3 was more enriched in heavy isotopes than at the other

sites, with C3 having mean values of -49.2 and -5.4‰ for $\delta^2H$ and $\delta^{18}O$, respectively. This indicated a greater impact of

isotopically enriched sewage on groundwater under the TWR channel. The range of stable isotopes in C0, C1, and C3 sites

were larger than in C5 and C6, suggesting the groundwater at the south sites were more easily affected by changing surface

condition than the north sites.

The residence time of groundwater of each site was calculated using Eq. (1-3) and presented in Table 1. There was large discrepancy between results calculated by δ²H and δ¹⁸O. Since δ¹⁸O is more affected by evaporation fractionation (Filippini et al., 2015), the δ²H results were used to investigate the residence and response times to precipitation or irrigation signals. The residence time for groundwater from C6 to C0 decreased from 382 to 97 days, similarity, the response time for groundwater from C6 to C0 increased from 266 to 118 days.

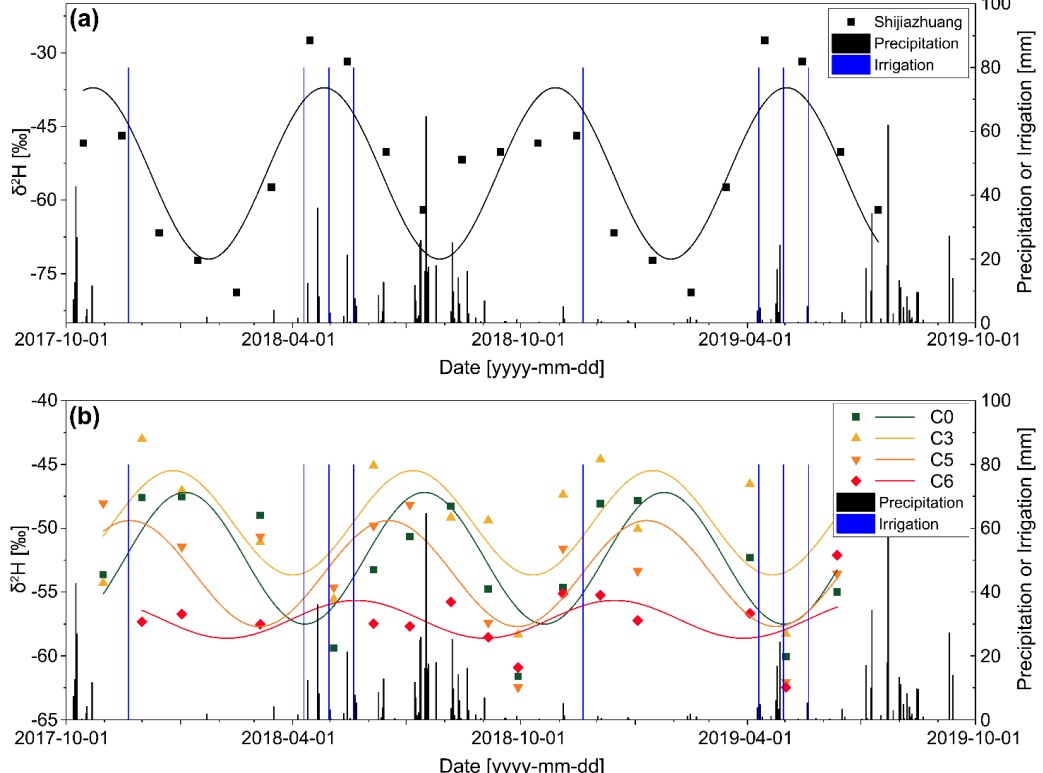

**Figure 7.** Variations of δ²H in rainwater **(a)** and groundwater**(b)** from November 2017 to June 2019. C0 was not shown in this figure due to the bad fitting coefficient.

The relationships for δ²H and δ¹⁸O in groundwater for the five sites (C0, C1, C3, C5 and C6) are presented in Fig. 8. Most groundwater samples plotted beneath the LMWL and along the TWR evaporation line. As shallow groundwater from 80-120 m depths are used in agricultural irrigation, it can be referred as an end member (Shallow GW). Most groundwater samples from C3 are in the upper boundary of the green triangle of the mixing plot (pink shadow region), plotting between mean precipitation and the most isotopically enriched porewater at shallow depths in the C3 profile as well as overlapping the sewage samples at the transect of the study area collected in 2008, 2009, and 2011. Thus, the groundwater at C3 is a mixture of summer precipitation and sewage. Most samples from C6 are distributed in the lower boundary of the mixing plot between the groundwater and shallow pore water end-members. This suggests GW and shallow porewater impact groundwater (at 30 m





depth) at this site. The C0, C1, and C5 samples all plot along the upper boundaries of the mixing space, suggesting the dominant

impact of precipitation and some legacy wastewater in the soil.

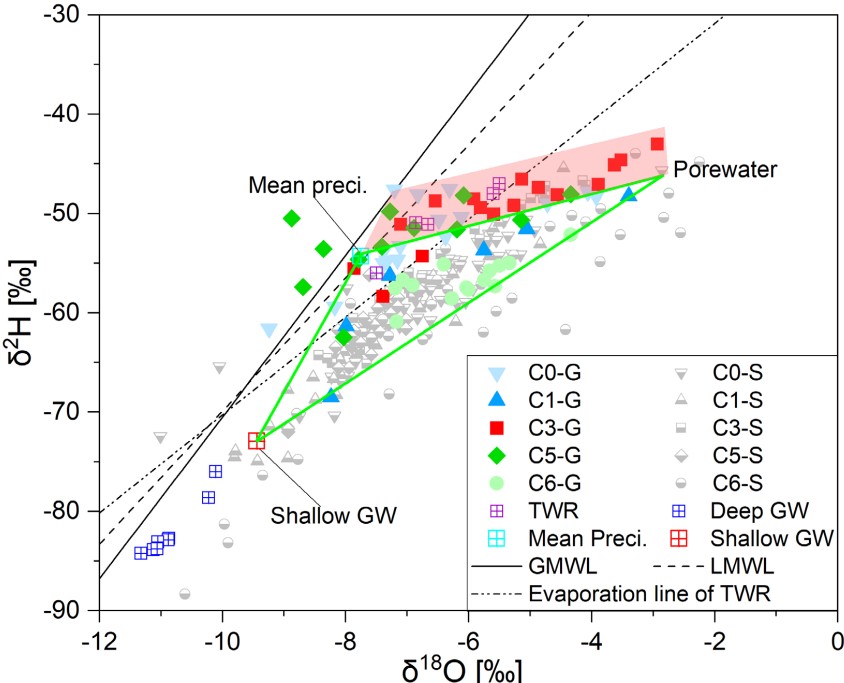

**Figure 8.** Relationship of $\delta^2H$ and $\delta^{18}O$ in groundwater (at 30 m depth) in C0-C6. Shallow GW means samples were collected in shallow

groundwater with depth of 100-120 m. Deep GW means samples collected in deep groundwater with depth ranging from 120 m-300 m.

Shallow GW, Mean Preci., and Porewater means three end-members of groundwater used for irrigation, mean precipitation used in mixing

calculation, respectively. G after the number of sites C0-C6 means groundwater, while S after the number of sites C0-C6 means porewater.

**Table 1.** Mean response times (Δt) and residence times (T) and corresponding fitting parameters of precipitation and groundwater using the

trigonometric function. Note: C1 was not shown in this table due to the large fitting error.

| Sites | $^2H$ R² | $^{18}O$ R² | $^2H$ $\delta_0$ (‰) | SE | $^{18}O$ $\delta_0$ (‰) | SE | $^2H$ $\delta_A$ (‰) | SE | $^{18}O$ $\delta_A$ (‰) | SE | $^2H$ Δt (d) | SE | $^{18}O$ Δt (d) | SE | $^2H$ T (d) | SE | $^{18}O$ T (d) | SE |
|---|---|---|---|---|---|---|---|---|---|---|---|---|---|---|---|---|---|---|
| C0 | 0.56 | 0.62 | -52.35 | 0.9 | -6.47 | 0.30 | 5.16 | 1.32 | 1.55 | 0.43 | 118 | 13 | 143 | 11 | 97 | 2 | 34 | 1 |
| C3 | 0.34 | 0.47 | -49.58 | 1.2 | -5.54 | 0.39 | 4.09 | 1.74 | 1.81 | 0.58 | 100 | 21 | 108 | 17 | 125 | 3 | 28 | 1 |
| C5 | 0.47 | 0.13 | -53.55 | 1.29 | -6.97 | 0.45 | 4.15 | 1.56 | 0.57 | 0.53 | 85 | 28 | 72 | 103 | 133 | 4 | 149 | 7 |
| C6 | 0.19 | 0.44 | -57.14 | 0.81 | -6.35 | 0.22 | 1.48 | 0.98 | 0.74 | 0.27 | 266 | 29 | 102 | 17 | 382 | 13 | 91 | 4 |
| GNIP shijiazhuang | 0.70 | 0.58 | -52.64 | 2.85 | -7.69 | 0.53 | 17.19 | 3.96 | 2.41 | 0.72 | - | - | - | - | - | - | - | - |



The contribution of different recharge sources can be estimated by end-members calculated using Eq. (4)-(6) using the three end-members: groundwater for agricultural irrigation (Shallow GW), weighted mean precipitation (Mean preci.), and evaporated porewater (Porewater) in soil as shown in Figure 8. The end-member of evaporated porewater represents the remaining wastewater. We calculated the seasonal variation of contribution from different recharge sources as shown in Fig. 9. The contribution of porewater at C3 (mean 44.0%) was larger than C6 (mean 38.0%), C0 (mean 28.5%), C1 (mean 26.7%)

and C5 sites (17.6%) corresponding with the longest residence time (Table 1). The contribution ratio of irrigation water at C6 (mean 31.7%) was also larger than at other sites, which might be attributed to lateral recharge as C6 is located in the upper stream of the transect. For sites of C5, C3, C1, and C0, precipitation also contributed greatly to groundwater recharge though groundwater response lagged.

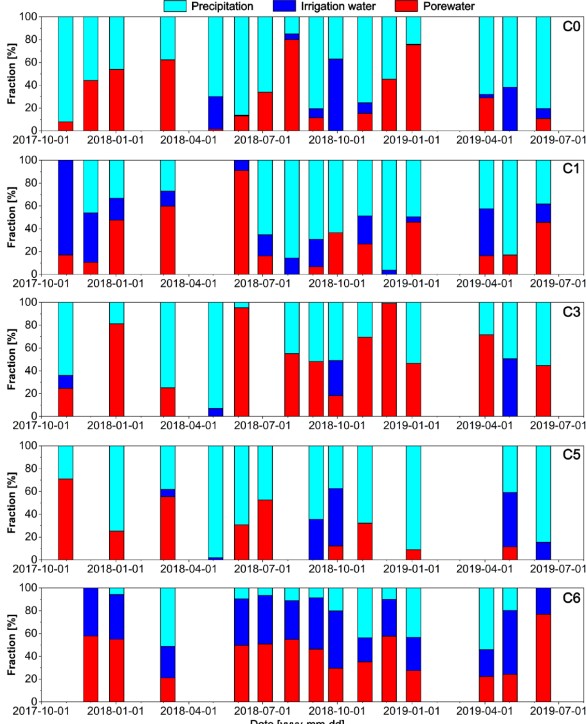

**Figure 9.** Seasonal variation of contribution ratios of precipitation, porewater, and irrigation water for sites C0-C6 based on isotope endmember mixing.

### 3.5 Water chemicals in groundwater

Figure 10 shows the monitoring results for indicative ions of $SO_4^{2-}$ and $NO_3^-$ concentrations in groundwater affected by wastewater. The $SO_4^{2-}$ concentrations tended to decrease (Fig. 10a) over the monitoring period. Mean $SO_4^{2-}$ concentrations

decreased over time with a maximum and minimum values of 1420.9 mg/L and 496.1 mg/L, respectively. Sulphate concentrations were slightly higher during the dry season than the wet season. Spatially, the $SO_4^{2-}$ concentrations at C3 and C5

sites were higher than at farmland sites C0 and C6 before June 2018 when sewage in the TWR was cleaned. The variation in concentrations for these sites reversed after that time. However, $NO_3^-$ concentrations differed significantly, showing a sudden increase and decrease in response to precipitation and irrigation, with a maximum value of 40.8 mg/L (Fig. 10b). The high

concentration peaks were observed in June or July after intense rainfall, and in May or June after spring irrigation of farmland, and there is also a slightly increase in $NO_3^-$ from January to March. The mean $NO_3^-$ concentrations in groundwater at different sites were related to the land surface conditions, with the highest value at the TWR channel (C3), the second highest value in wheat/maize crop at C0, and the third highest value in corn crops (C6), and the lowest value at the two sides of the TWR embankment (C1 and C5).

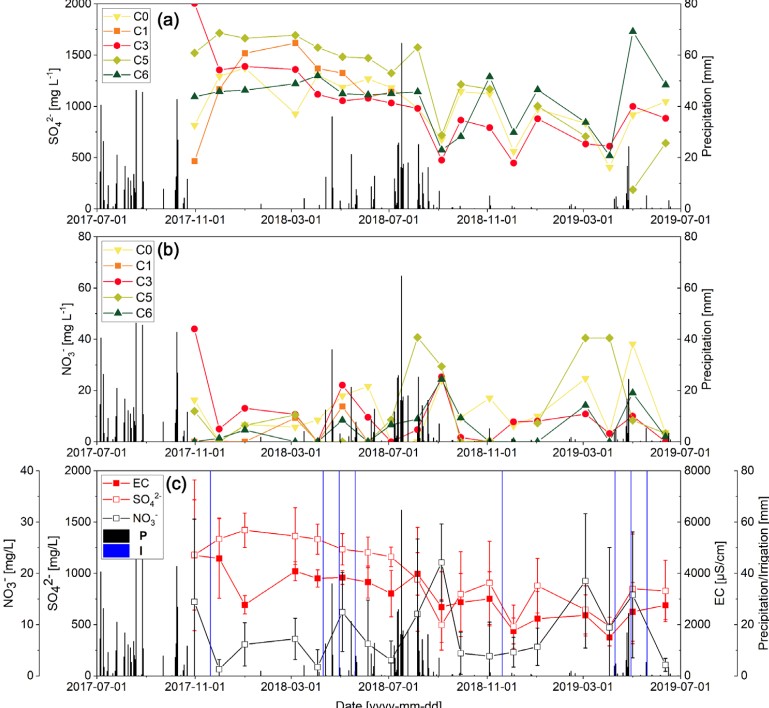

**Figure 10.** Variations of major ions in groundwater in boreholes along the section from November 2017 to June 2019. **(a)** $SO_4^{2-}$; **(b)** $NO_3^-$; **(c)** mean values for EC, $SO_4^{2-}$ and $NO_3^-$ of all boreholes (the error bars indicate the ranges).

## 4 Discussions

### 4.1 Infiltration processes affected by multiple recharge sources

As Fig. 5, and 6 show, the stable isotopes in the five soil profiles indicated differences in infiltration processes and water sources in porewater when sewage remained in the TWR channel. All vertical profiles have an obvious isotopic peak point for indicating the vertical variation in sources. These points were 900, 450, 650, 875 and 750 cm for C6, C5, C3, C1, and C0,

365





respectively (Fig. 5), suggesting infiltration and mixing of different sources. For example, the C3 profile contained sewage from the TWR and the enrichment of $\delta^2$H and $\delta^{18}$O from deep to shallow depths indicates infiltration and mixing of the more

enriched sewage in the TWR after 2011 with previously recharged sewage (Fig. 3). The enrichment of stable isotopes in deep layers (>650 cm depth) and the quickly enriched isotopes in shallow layer (<650 cm depth) of the C3 profile were in accordance with the stable isotopes in the sewage of the TWR before 2011 and the greater enrichment from 2011 to 2017, respectively. According to the recharge depths (650 cm) and the recharge time of 6 years from 2011 to 2017, the average recharge rate of sewage was estimated to be 109 cm/year for C3. Similarly, the vertical distributions of isotopes in the farmlands with the C0

and C6 profiles revealed the enriched sewage that was irrigation before 2005 and shallow groundwater irrigation with depleted isotopes (mean -62.5 and -7.9‰ for $\delta^2$H and $\delta^{18}$O, respectively) from 2005 to 2017. According to the recharge depths of 750 and 900 cm and recharge time of 12 years for C0 and C6, respectively, we estimated recharge rates of 63 and 75 cm/year for C0 and C6, respectively. These recharge rates are in the range of 41 to 200 cm/year as estimated by previous research completed in the alluvium plain of the North China Plain (Min et al., 2018; Wang et al., 2019). Due to the high soil moisture under sewage

(Fig. 5), the recharge rate of C3 was higher than the cropped fields of C0 and C6. The relatively greater recharge rate in the cotton field of C6 compared to the wheat/maize field of C0 can be attributed to the relatively thinner silt layer in C6. Profiles C1 and C5 were located outside of the TWR embankment. A T test showed that isotopes in the C1 and C5 profiles were similar to that of C3. This indicated that the lateral flow from sewage in C3 affected the stable isotopes of the porewater at depths greater than 450 and 875 cm for C5 and C1, respectively. This was verified by a field survey that found

the polluted sludge distributed along the silt clay layer under this depth, which also supported lateral flow of sewage (Fig. 3a).

The infiltration and mixing processes are very similar to previous research conducted in the deep vadose zone when affected by an unlined wastewater pond (Asaf et al. 2004, Gal et al. 2009). Gal et al. (2009) found that the recharge rate decreased after the wastewater discharge ceased due to the decreasing soil moisture. Additionally, variation of surface conditions resulted in

different distributions of isotope enrichment at vertical depths (Gal et al. 2009). Most $\delta^2$H isotopes in the deep layers of C6 and C0 were greater than the average $\delta^2$H (-49.7‰) in sewage from September 2008 to June 2011. During extensive evaporation from the unsaturated zone, kinetic effects by vapor diffusion were greater than those associated with evaporation from open surfaces (Clark and Fritz, 1997). Therefore, stable isotopes throughout the C6 and C0 profiles had greater enrichment than sewage in TWR and porewater in C3. When comparing C6 and C0, the fractionation effect from soil

evaporation beneath cotton crops was stronger than for soils beneath wheat/maize, with the slopes of the evaporation lines being 2.60 and 4.13, respectively (Fig. 6b). This can be attributed to differences in water balances in the different land uses. The cotton field receives 100 mm of irrigation a year (Min et al., 2018) and wheat/maize cropping system receives 320-360 mm of irrigation a year (Sun et al., 2006; Zhang et al., 2018). The dryer soil conditions resulted in stronger evaporation fractionation, while more irrigation at the wheat/maize fields diluted the isotopic fractionation signal.



**4.2 Seasonal groundwater recharge and groundwater flowpaths**

The seasonal recharge of input sources is demonstrated in the seasonal variation of the stable isotopes in the groundwater (Fig. 11). The lag time or response time ($\Delta t$) of groundwater to precipitation/irrigation ranges from 85 to 266 days in C0 to C6 (Table 1), with a mean of 131 days. Benettin et al., (2018) used numerical experiments to model the isotopic evolution of seasonally varying precipitation inputs under the influence of seasonally varying evaporation. They found that if the seasonal cycle of evaporative fractionation is not in phase with the seasonal cycle in source water composition, the residual water samples will trace out a hysteresis loop. Different from other sine or cosine curves with a period of one year in precipitation and groundwater (Lee et al., 2007; Tekleab et al., 2014; Benettin et al., 2018), the seasonal variation of precipitation and groundwater showed features of semi-annual periodic sine or cosine curves due to the impact of seasonal irrigation in our study area (Yamanaka et al., 2004). Fig. 11 shows both the observed $\delta^2$H vs. $\delta^{18}$O relationship of pore- and groundwaters and the monthly values for the sine wave fitting curves for each site. It displayed the distribution of two loops with a mean hysteresis time of three months. The contribution of each of the seasonal recharge sources determines the shape of the loop. As for groundwater at C3 influenced by TWR sewage, the loop is narrow with a nearly straight line of precipitation and evaporated porewater end-members crossed the TWR evaporation line. Samples during summer and winter overlapped with porewaters at shallow depths, suggesting a major contribution of isotopically fractionated porewater on groundwater at those times while samples during fall and spring suggest a dominant impact of (non-fractionated) precipitation. Porewater contribution to groundwater recharge occurred during all seasons at C6, but the contribution of precipitation and shallow groundwater increased at C0 site, which resulted in larger loops at the C0 site compared to the C6 site. Groundwater in irrigated farmlands with low amounts of irrigation and strong evaporation at C6 (cotton lands) had a concentrated loop that overlapped with shallow porewater suggesting a major contribution of porewater.



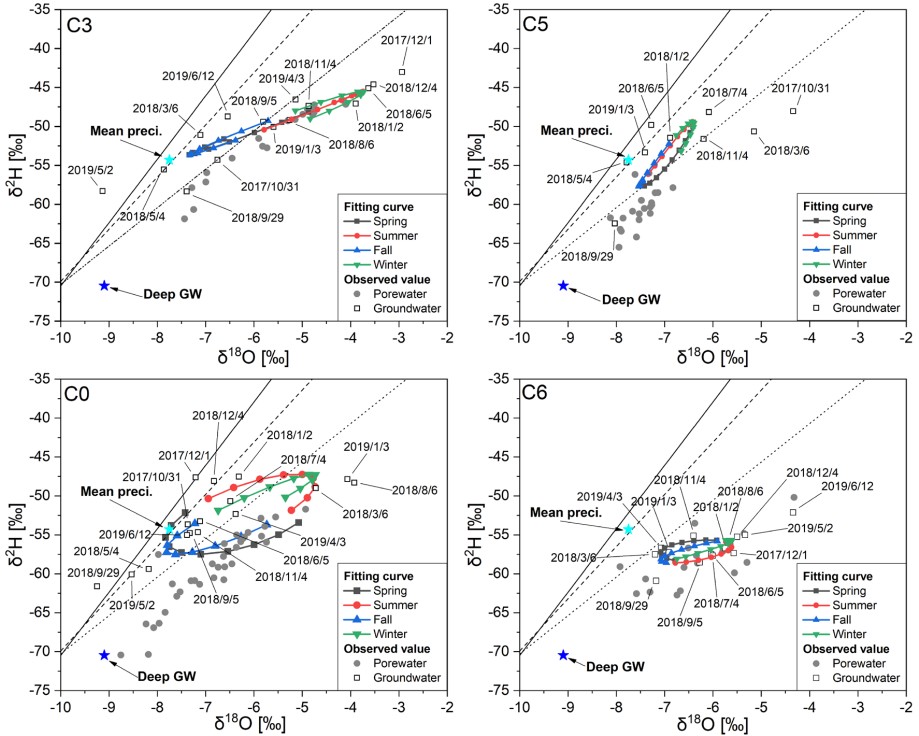

**Figure 11.** Relationship of $\delta^2H$ and $\delta^{18}O$ in groundwater of C0-C6 compared with fitting sine/cosine curves of different seasons.

According to the contribution ratios of different sources (Fig. 10), we found that precipitation also contributed greatly to

groundwater recharge beside the residual water in the porewater. However, according to the estimation of recharge rates of

translatory flow (65~109 cm) from the distribution of vertical isotopes, it is impossible for surface water sources to reach the

water table in one year. Compared with the recharge rates calculated by porewater, the response time indicates that there must

have been preferential flow or fast flow to the groundwater. The abrupt changes in porewater isotopes in the sand aquifers (Fig.

5) suggest that preferential or fast flow (such as lateral groundwater recharge) included evaporated water sources. Many studies

conducted on soil water or groundwater in the North China Plain have supported the existence of preferential flow or fast flow,

particularly for irrigated farmlands (Song et al., 2009; Ma et al., 2017; Zheng et al., 2020). According to the field survey,

preferential flow traced by pollutants was also found in silt layer and silt clay above 20 m depth (Fig. 3b-d). Therefore, it may

be possible that preferential flow contributes to seasonal variation in groundwater isotopic signals. Unlike C6, C0-C5 are

located in the Tanghe River Channel, and the higher permeability of the preferential flow in the unsaturated zone possesses a

shorter mean groundwater response time of 100-118 d.Eventually the groundwater affects the aquifers in the study area by

lateral flow. To facilitate understanding, we synthesized the above discussion of groundwater recharge and drew a conceptual

diagram of TWR upper shallow groundwater recharge model (Fig. 12).





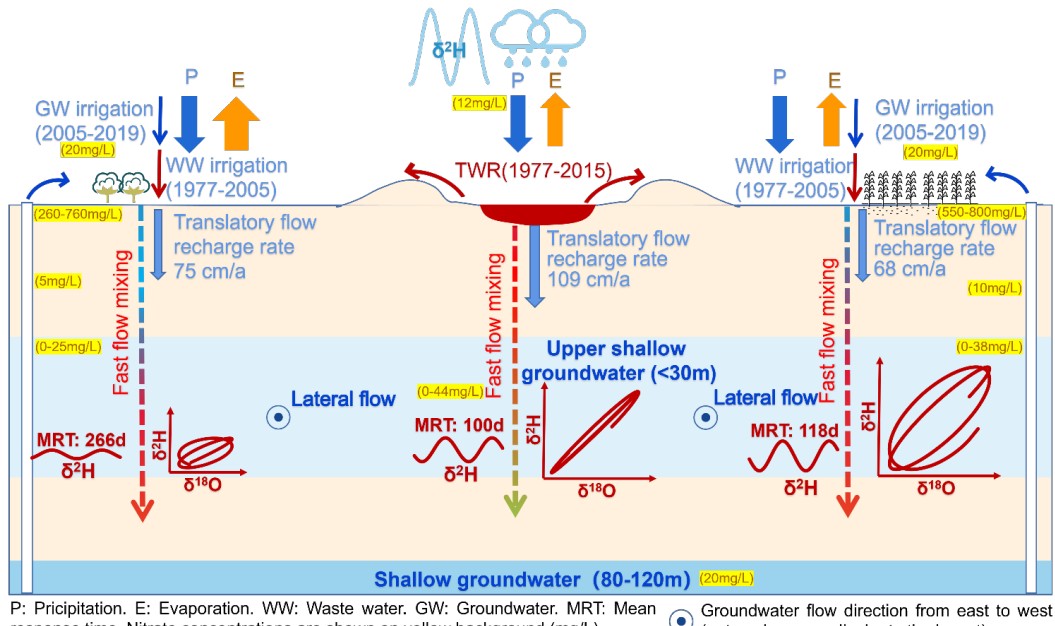

**Figure 12.** Conceptual model of groundwater recharge and transport of nitrate affected by multiple sources with seasonal variation.

### 4.3 Impact of preferential flow or lateral flow on groundwater quality

Understanding the processes of local groundwater recharge is important to identify the solute transport in soil water and contamination in groundwater. Besides stable isotopes, water chemicals in groundwater are good indicators for verifying groundwater processes. Sulphate was the major water ion in sewage of the TWR which contributed to variation in water quality (as EC) (Wang et al., 2014). As Fig. 10 shows, the $SO_4^{2-}$ were relatively stable due to the infiltration of sewage with high $SO_4^{2-}$ concentrations (Fig. 4) before June 2018 and there were large variations after the 2018 rainy season since the $SO_4^{2-}$ source was cut off. The dilution effect caused by preferential flow or fast flows of precipitation recharge might be the primary cause of this variation. This is in accordance with the contribution ratio of precipitation (Fig. 9). However, the increased $SO_4^{2-}$ concentrations after fast flow recharge indicated the contribution of translatory flow or remobilization of porewater which had a high $SO_4^{2-}$ content (Fig. S3). Similarly, winter and spring irrigation with shallow GW had a similar impact on groundwater quality.

The impact of preferential flow and slow translatory flow on $NO_3^-$ in groundwater is different from $SO_4^{2-}$. A conceptual model of recharge mechanisms of groundwater and transportation of pollutants (i.e., $NO_3^-$) is presented in Fig. 12. During the infiltration processes of multiple water sources, evaporation, mixing, and recharge rate of translatory flow determine the vertical distribution of isotopes in porewater. Isotopes in aquifers are different from upper layer porewater, but are similar to shallow depth porewater, suggesting the mixing of fast flow. We found that the fast flow contributed in great part to the



groundwater recharge and increase in $NO_3^-$. Past research has pointed out that $NO_3^-$ accumulates in soils to 2 m depth (Chun-Sheng et al., 2011; Ju and Zhang, 2017; Wang et al., 2019) and the amount accumulated decreases with increased depth toward the unsaturated zone. Generally, $NO_3^-$ accumulates during the winter wheat season and leaches into the soil profile during summer maize season (Zheng et al., 2019). In our study, leaching occurred not only in summer, but also in the winter, showing

elements of seasonal leaching. This can be attributed to a number of factors. First, the fast flow induced by flood irrigation carrying the N-fertilizer in surface soil into aquifers. Second, there were many factories around the TWR and these operations dug seepage pits to store or leak sewage, as was noted during our field investigations. Farmland was the major land use type in the study area and also on the plains of the Baiyangdian watershed. It is easy for $NO_3^-$ to be transported from the root zone to shallow groundwater in farmland. Min et al. (2018) reported that $NO_3^-$ concentrations in peaks above 2 m ranged from 550-

812 mg/L for a wheat/maize cropping system and ranged from 260-760 mg/L for cotton, and differences between crops were due to different N-application levels in the North China Plain. This could also explain why the $NO_3^-$ concentrations in the groundwater of the wheat/maize crop (C0) were higher than that in groundwater of cotton (C6).

**5 Conclusions**

We investigated recharge and transport pathways of major pollutants ($SO_4^{2-}$ and $NO_3^-$) from the surface to the upper shallow

groundwater in an alluvium river aquifer as affected by a long-term unlined wastewater reservoir (the TWR). The variation in contributions from many sources created a complex groundwater recharge system. As the TWR was built overlying an alluvium river aquifer, the long-term wastewater leakage and irrigation resulted in the deterioration of groundwater quality around the wastewater reservoir.

The stable isotopes in porewater of soil profiles record the infiltration and mixing processes of different water sources including

480 precipitation, leaked sewage, sewage irrigation, and shallow groundwater irrigation. The average vertical recharge rate of translatory flow ranged from 63 to 109 cm/year. On farmlands, the effect of evaporation fractionation on porewater isotopes in cotton fields was greater than in wheat/maize the cropping system. The stable isotopes in sand aquifers were similar to those in porewater at shallow depths in the soil profiles, suggesting the existence of fast preferential flow pathways.

Shallow groundwater above 30 m showed seasonal contribution of evaporated water sources including precipitation, porewater

in soil, and irrigation water from shallow groundwater from 80-120 m depths. The preferential flow pathways and lateral flow along the alluviums river channels, lead to large contributions to groundwater recharge from precipitation with mean values ranging from 44 to 61% for most sites. Our research revealed the distribution of two hysteresis loops for groundwater samples in the $\delta^2H$ and $\delta^{18}O$ plot. The possibility of preferential flow recharge increased the contribution of precipitation and increased the uncertainties of the residence time. However, the hysteresis loop of seasonal groundwater recharge was affected by seasonal

variation in water sources, and this variation was consistent with isotopes signals between groundwater and different sources.



We found two types of hysteresis loops in the $\delta^2H$ and $\delta^{18}O$ plot of groundwater: 1) groundwater in regions influenced by TWR sewage had narrow loops and a nearly straight line of precipitation and evaporated porewater end-members crossed the TWR evaporation line; and 2) groundwater in irrigated farmlands with low and high amounts of irrigation and strong evaporation (i.e. cotton fields and wheat/maize fields) had a concentrated loop that overlapped with shallow porewater suggesting all three water sources made contributions. The deviation from the LMWL showed differences in the magnitude of evaporation. Therefore, it was revealed that the shape of the hysteresis loop in the $\delta^2H$ and $\delta^{18}O$ plot of groundwater was determined by the extent of different recharge sources of seasonal recharge, which is also the novelty of this research.

The seasonal recharge also contributed to the seasonal variation of groundwater quality. The $SO_4^{2-}$ concentrations slightly decreased due to dilution. However, the $NO_3^-$ concentrations increased after precipitation and irrigation when preferential flow carried nutrients from the shallow soil depths into the aquifer. Therefore, local recharge and soil pollutants were affected by long-term wastewater leakage and irrigation, and these measures should receive greater attention. Action is needed to prevent pollutant transport by preferential and fast flow in river alluvium aquifers as these systems are highly susceptible to contamination.

## Acknowledgments

The study was supported by the National Natural Science Foundation of China (No.42071053), the National Key R&D Program of China (2021YFD1700500) and the Foundation for Innovative Research Groups of the Natural Science Foundation of Hebei Province (D2021503001).

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
