# Peer review of "Seasonal recharge mechanism of the upper shallow groundwater in a long-term wastewater leakage and irrigation region of a river alluvium aquifer"

_Hydrology and Earth System Sciences, 2022_

## Author Comment (AC1)

Thank you for your careful reading, many of the suggestions have been very helpful to us. Below are our replies and modifications based on your questions and suggestions. " **RC1**" refers to the remakes of referee 1 and "**Reply**" refers to our answers. In the "**Reply**", we used the present perfect tense to express the changes we made based on the reviewer's comments. And, as needed, we have italicized the changed text.

**GENERAL REMARKS**

**RC1:** The paper addresses an interesting issue based on the interpretation of isotopic and hydrochemical tracers. Through all the paper the horizontal and vertical variations of all the considered parameters are shown, interpreted and discussed but the space-time distribution of these changes is never related to the sedimentary architecture of the subsurface. This is a relevant point to be addressed by the Authors in my opinion, because all the recharge, mixing and exchange processes in the subsurface are controlled also by textural, structural and compositional heterogeneity and anisotropy of the porous medium. See the following specific comments and suggestions.

**Reply:** In the original manuscript, there is little description of the heterogeneity and anisotropy of the sediment texture, structure and composition, and we will add the corresponding description in the revised manuscript.

**RC1:** Some parts of the paper are difficult to read, because of the misleading use of some terms and unclear sentences. Some relevant information is missing, some is given almost at the end of the paper. A complete and thorough characterization of the regional setting and of the specific features of the very small study site should be addressed to a dedicated section at the beginning of the paper. All the collected samples and the other sources of data should be mentioned in the Methods section. See the specific remarks on these points.

**Reply:** Combined with your specific suggestions, we have made careful modifications in following reply to your specific comments and suggestions within the full text.

**RC1:** Some citations in the main text are not complete (e.g. Zhang, 2009 should correspond to Zhang et al., 2009? Please note also that this is the only reference to a paper that plausibly might contain some information on the setting and subsurface stratigraphy of the region, but it is impossible to find out). I did not cross-check the citations vs. reference list.

**Reply:** We have checked again and this is the only incorrect citation in the whole manuscript, which has been changed to (Zhang et al., 2009).

**RC1:** Several language and typewriting mistakes occur through the main text and the figure captions, some sentences are difficult to read. I did not mark them; the Authors will check by themselves. A thorough revision of the English form is suggested.

**Reply:** We've checked throughout the text. We believe that the language has been improved after the throughfall checking.

**SPECIFIC REMARKS**

**RC1:** Page 4, line 110-114: what do you mean when You state that the south embankment of the TWR was built "in the Tanghe River channel"? Was it already abandoned by the river? When and why did abandonment occur? Which is Your use of the term "embankment": man-made bank or also natural levee? In Fig. 3 the N and S embankments are drawn as natural silty layers, that You consider as horizon I for sampling. Is it true? What did You sample, sediments or artifacts? Which are the relationships between natural sediments and manmade embankments?

**Reply:** The sentence means that the Tanghe Wastewater Reservoir (TWR) was built within the original Tanghe River channel and the south embankment was man-made and it was used to store wastewater since 1977. The Tanghe River channel has been drying up since several decades due to the interception of the reservoir upstream in the mountains and the over-pumping of groundwater in the plain area. In addition, the north embankments of the TWR was also built by man-made as the flood controlling embankments. Both embankments are about 180m apart. In the original Fig. 3, the two embankment's misused the pattern of silty, in fact they are both compacted and stacked using the silty sediments from the Tanghe river channel. Finally, we modified Figure 3.

[Figure]

*Fig. R1   Modified Figure 3.*

**RC1:** Page 4, line 116: a section should be added to describe the geomorphology, hydrology and subsurface stratigraphy of the site. When was the Tanghe River channel dried? Was it a natural or manmade process? Which kind of lake is Baiyangdian Lake? A composition of tens of ponds in a wetland or a deep lake with internal circulation? How does it feed the free aquifers? How did it evolve to its present-day condition of water shortage? Was a river mouth built by the Tanghe river into the Baiyangdian Lake? Which kind of mouth, a delta? Were the sediments that You studied deposited in delta-plain/delta-front or in palustrine settings? Was the Tanghe River a stable anastomosed river with stable islands and overbank areas? Which is the influence of the present-day elevation above sea-level (in Your cross-section elevation of the riverbed is about 6 m a.s.l. and You state at line 215 that the elevation of the transect is 8-10 m a.s.l.)? Can You exclude any tidal influence on sedimentation during Quaternary? Are the sediments You are sampling Holocene to recent in age or older? The setting of Your very small study area is relevant under different viewpoints: it is the starting point to reconstruct heterogeneity of the shallow subsurface and the

reader needs this info to understand the results You show, for instance in Fig.3. Considering the shallow groundwater gradient and flow lines, a significant reversal should be occurred since the time the Tanghe river was flowing into the lake and building the sedimentary bodies, up to the present-day configuration in which the lake feeds the adjacent dried alluvial and/or deltaic plain. If I understood Fig.1 (please complete the caption: does the purple frame in c correspond to box d?), the study site is about 8 km west of the present-day lake shore, is it correct? The setting is relevant also because You conclude with some general considerations on seasonal recharge mechanisms deduced from Your very small site (a 2D strip 400 m long), without considering sediments heterogeneity and the general setting at all.

**Reply:** Good suggestions. We added some descriptions of geomorphology, hydrology and subsurface stratigraphy in section 2.1. They are italicized as follows, where the bolded parts are the main additions. In addition, Baiyangdian Lake is a wetland type lake consisting of many small lakes. The Tanghe River mouth has no delta or tidal influence and it is just an inflowing river into an inland lake. Our study site was located at the middle of the TWR, 9.5km from Baiyangdian Lake. The sediments are mainly characterized by floodplain deposition. Moreover, the purple box in Fig. 1c is consistent with the range in Fig. 1d, which we have marked.

*2.1 Study site*

*Our study site is **located at the middle of** the Tanghe Wastewater Reservoir (TWR) in the Xiong'an New Area of the Baiyangdian Lake watershed, **9.5km far from Baiyangdian Lake** (Fig. 1a). The site is characterized by a continental monsoon climate with a strong seasonality. The weather is cold in the winter and warm in summer with an annual mean temperature of 13 °C. The annual mean precipitation is 504 mm/year, of which more than 75% occurs between July and September (rainy season) as heavy rainfall events (Fig. 1b). The potential evapotranspiration is 1208 mm/year and the evaporation from March to June is higher than during other months.*

*The reservoir is unlined and was built **along the northern part of Tanghe river channel** (Fig. 1c) for storing untreated industrial sewage with high concentrations of sulphate from Baoding City for almost 38 years (from 1977 to 2015). **It was connected to the lake through a sluice gate in the east, affecting the lake's water quality severely**. The TWR has a total length of 17.5 km, width of about 100 m water surface, and a storage capacity of $7.2 \times 10^6 m^3$. The north embankment of the TWR was **the original Tanghe River embankment, constructed in 1966**, while the south embankment is **compacted from the silty sediment excavated from TWR inside the river channel, both embankments are about 180m apart. The Tanghe River has been drying up since several decades due to the interception of the reservoir upstream in the mountains and the over-pumping of groundwater in the plain area, and a large amount of wheat/maize is planted in the channel.** After the TWR started storing sewage in 1977, the impacts of anthropogenic activities on the TWR varied greatly (Fig. 2). The amount of sewage entering the reservoir was reduced and stopped in the middle part of the channel in 2010. The sewage inflow was totally intercepted in 2015. Due to the development of the Xiong'an new area in 2017, the sewage that remained in the reservoir was removed in 2018.*

***TWR is located in the floodplain with flat terrain, and the surface elevation of the profile at the study location is 6-10 m. According to the regional hydrogeological survey, the 30 m depth range consists of Quaternary Holocene and Late Pleistocene alluvial sediments with a high level of permeability and heterogeneity, in which the unsaturated zone interlayered with silt and silty clay, and the saturated zone comprises silt and sand with local inclusions of thick layers of silty clay.***

*During the sewage storage period of TWR, the upper shallow groundwater flows from north to south, and the groundwater level inside the reservoir is higher than that of the outside of embankments (Wang et al., 2013)*. At a regional scale, over-pumping of groundwater has led to the drying of natural rivers and lakes, and the decline of groundwater levels. However, artificial water recharge to the Baiyangdian lake increased the water table around the lake, the groundwater level in the area around TWR shows a high level in the east and a low level in the west (Yuan et al., 2017; Yuan et al., 2020). Groundwater flows from northeast to southwest with a hydraulic gradient of 0.5‰-1.3‰ at the middle of TWR in Fig. 1c (by kriging interpolation with the groundwater level data measured in November 2018). **The average groundwater depth along the profile is 12.7m, and the permeability of the phreatic aquifer is 4.66m/d.** Winter wheat and summer maize are the major land uses around the TWR (Fig. 1 c and d). The wastewater from the TWR was mainly used for agricultural irrigation during the past several decades. In recent years, shallow groundwater (80-120 m depth) was used for irrigation. Along with the irrigation return flow of sewage and shallow groundwater, infiltration of wastewater is another major source for groundwater along the TWR. It has been estimated that 76% of the sewage recharges into an aquifer by leakage and irrigation and the influencing range was about 3 km around the TWR and 150 m depth of the aquifer (Wang et al., 2014).

[Figure]

*Fig. R2    Modified Figure 1.*

**RC1:** Page 6 line 151: what do You mean by "soil"? This term might be restricted to the complex stratigraphy of pedogenetic profiles. You state that samples were collected along a 30 m deep soil profile. Is this the thickness of the weathering profile? If so this would be extremely relevant and the horizons of soil stratigraphy should be carefully described. Otherwise You are talking about alluvial/lacustrine sediments stratigraphy, including eventual intervening soil and paleo-soil profiles. It should be clarified. In the second hypothesis, the term soil should be replaced with "sediments". This is not just a boring terminology argument, considering the different hydrogeological and hydro-chemical behavior of a 30 m thick composite soil profile compared to a stratigraphic column of alluvial plain, palustrine, deltaic, lacustrine (??) sediments and eventual intervening soils and

paleosoils. At last, in Your discussion section You differentiate soil water from groundwater, highlighting the need of a proper use of the term soil in aquifer characterization. Surface soil, shallow and deep unsaturated/vadose zone or soil, shallow vs. deep aquifers or groundwaters are recurrent terms through Your text, often used in a contrasting way, that is misleading. Please check the proper and consistent use of these terms from the introduction to the conclusion sections.

**Reply:** The term 'soil' should be replaced by 'soil-sediment'. For the root zone less than 2m, it can be termed as soil. However, for deeper alluvial layers it is more appropriate to call it sediment. Regarding the part of hydrogen and oxygen isotope description of pore water, considering the sediment profile depth of 30m, including unsaturated and saturated zones, the description with depth information is used to visualize    the profile. Moreover, regarding the upper shallow groundwater, shallow groundwater and deep groundwater, we have added descriptions in section 2.1 to facilitate readers to distinguish them. In the study location, the Tertiary and Quaternary aquifers are usually divided into shallow groundwater (aquifers I and II) and deep groundwater (aquifers III and IV), however, our study is on groundwater less than 30 m deep, which we called as the upper shallow groundwater for distinction (Fig. R3). The shallow groundwater (80-120m) mentioned in the text refers to the main groundwater irrigation source, while the deep groundwater is only mentioned when discussing the recharge source. A figure to describe the shallow groundwater and deep groundwater has been added to the Supplement as Fig. S1 placed in front of the original Fig. S1.

[Figure]

*Fig. R3    Quaternary hydrogeological profile of the study area (Fig. S1).*

**RC1:** Section 3.1: how did You obtain the geological cross-section in Fig.3? How did You correlate the data from the boreholes? Which are the sediments involved? What do they represent? How many sediment packages could You recognize in each borehole? Are there erosion surfaces? Are there soils between them? The photos in Fig.3 show clear pedogenetic features in the silty-clay and silt sediments. How many superimposed weathering profiles could You observe? Might they help the lateral correlation between boreholes? Could You recognize any stacking pattern in the stratigraphy of Your borehole logs? May the occurrence of fining upwards sequences help the lateral correlation? How do You explain the convex-up top and flat base of the sand-bodies and the opposite concaveup shape of the silt and silty-clay units? How do You justify the curved shape of the lower sand-body/aquifer? Why are You sure that the lowermost sand layer in C5 correlates with the lowermost sand horizon in C4 whose top lays at least 5 m below the base of the sand layer in C5? Which depositional environment and stratigraphic dynamics would lead to such a configuration? Is this lower aquifer body really continuous in 2D? In Fig.5 You show large variations in isotope curves across the same sand layers: how do You explain this pattern? The descriptions in section 3.1 do not provide sufficient information about the 2D stratigraphic cross-section. In addition, I really do not understand the hydrostratigraphic meaning (if any) of horizons I, II and III (watch also the last sentence dealing with horizon III: is it complete?). Have You any data about the composition of sediments? The composition of the detrital and/or pedogenetic clays strongly impacts on groundwater and immiscible fluids chemistry, the fate of solutes and ions adsorption. At last You show 7 sampling sites but You present the data of 5 among them, discarding two sites within the TWR. Which is the reason for that?

**Reply:** The erosion surface and other signature interfaces, if observed, do help to map the cross-sectional hydrogeological profiles. However, due to the difference in professional background, we focused on the effect of different sediment textures on water and nitrogen transport. The two-dimensional profiles in the paper are mainly drawn by the spatial information of lithology and depth of the sediment profiles. The study is located in the downstream of the Tanghe River of the floodplain, where the topography is gentle and the river sediments have characteristics of spatial continuity, but show sedimentary layers of varying thicknesses spatially due to differences in depositional hydrodynamics and material carried. The study correlated seven sediment profiles based on their lithological characteristics, such as the presence of two layers of silty clay above the first sand layer, two sand layers, and a thicker silty clay layer below the base of the second sand layer. Finally, the connected 2D profiles can reveal roughly the depositional processes during the Upper Pleistocene and Holocene periods of the Quaternary. For example, the large difference in the spatial distribution of silt and silty clay between the first and second sand layers reflects the large difference in the spatial distribution of fluvial hydraulic conditions during that period. As for why the bottom of the first sand layer is so flat, the distribution range of the first sand layer shows that the river floods had a wide range of influence during the ancient river sediments, so it is possible that the river erosion was strong and long in the early period to form a flat erosion surface. In general, the distribution and morphology of the sedimentary layers are closely related to the surface hydraulic conditions and environment in the period of deposition.

Groups I, II, and III are to facilitate the characterization of water isotopes at depth, which is easily understood by moving it to "3.3 Stable isotopes in porewater".

In addition, we mentioned sediment sample collection only for C0/C1/C3/C5/C6 in section 2.2, of these sites, C2 and C4 are only used for observing sediment layering structure, not for analyzing the stable isotopes of porewater.

**RC1:** Section 3.2, lines 239 – 241: which is the source of the isotope data before 2015? The comparison with the data collected after the sewage inflow interruption is relevant, so the reliability of the source of isotopic data is important, insert references.

**Reply:** Thanks for your reminder. We have emphasized it again in the figure name.

*Figure 4. Variation of stable water isotopes and major water chemicals in sewage of the TWR. **All data are from our historical monitoring, among which data for 2008 and 2009 have been***

*published in previous works (Wang et al., 2014).*

**RC1:** Section 3.3. Here You discuss the stable isotope curves of pore waters almost completely disregarding the aquifer/aquitard heterogeneity. Even if Your 2D correlation scheme is questionable, the heterogeneity and anisotropy of the shallow subsurface sediments are apparent and might impact on flow paths and residence times of groundwater in the different sediment units. In Your comments on the three groups of samples You consider Group I samples altogether: which of them were collected on the TWR manmade embankment (at C1 and C5? You state that they were collected outside the embankments but in Fig.3 they are located on the embankments) and which in the natural sediments (if any) aside of them?

**Reply:** Considering that the section 3.3 belongs to the "Results" section, we present more the distribution and characteristics of the pore water isotopes, and the description and discussion of the heterogeneity and anisotropy of the unsaturated zone are described in the discussion section.

C1 and C5 are located on the outside of the TWR embankments, and Group I refers to the first layer of silt.

**RC1:** Line 256; what do You mean stating that "Variation in porewater isotope decreased with depth at all sites"? Excluding C3 I see large variations of both values at depth. The following sentences are also difficult to read.

**Reply:** There was a mistake with this sentence which has been deleted in the revised version of the manuscript. We described the variation in porewater isotopes in detail in the following sentences.

*The stable isotopes of different sites show gradual **enrichment** (C0, C1, C6) or **depletion** (C3) trend with depths due to the infiltration of different water sources. To identify the stable isotopes characteristics, the depth profiles were divided into three groups (Fig.5): (I) The top silt layer (above the first silty clay layer, where there was little evidence of the sewage leakage); (II) From the top of first silty clay layer to the top of first sand layer, where sewage might penetrate the silt and silty clay reaching the first aquifer; and (III) Depth from the top of first sand aquifer to the bottom of the second aquifer. The isotopic values then decreased at a relatively stable rate below 650 cm depth (Group III). The stable isotopes in the upper of Group II in the C3 profile (0-300 cm, excluding 60 cm) were most enriched and least variable with $\delta^2H$ ranging from -49.1 to -47.2‰ and $\delta^{18}O$ ranging from -5.2 to -4.1‰. Then the stable isotopes showed a depleting trend from 300 to 650 cm with peak values of -61.9 and -7.4‰ for $\delta^2H$ and $\delta^{18}O$, respectively. These variations are consistent with variations of the depleting isotopes in sewage in the TWR backward from 2017 (Fig. 4). Differing from the C3 profiles, the stable isotopes in C0 and C6 increased from the surface (Group I) with increasing depths (Group III), except for the silty clay depths deeper than 22.5 m. This is consistent with the increasing variation in isotopes from irrigation water with SGW at present (mean $\delta^2H$: -62.5‰; mean $\delta^{18}O$: -7.9‰) and with wastewater (mean $\delta^2H$: -46.0‰; mean $\delta^{18}O$: -5.4‰) historically. The ranges in isotopes in C6 (from -88.3 to -43.9‰ and from -10.6 to -2.25‰ for $\delta^2H$ and $\delta^{18}O$, respectively) were larger than in C0 (from -70.4 to -45.7‰ and from -8.8 to -2.9‰ for $\delta^2H$ and $\delta^{18}O$, respectively).*

[Figure]

**RC1:** Lines 284-285: I do not understand this sentence. *"There were few points in the aquifer where C1 showed extreme deviation from the TWR evaporation line (labeled in Fig. 6a)."*

**Reply:** The term "aquifer" was used inappropriately here, but we actually want to show that the anomaly of porewater isotopes in the sand layer, which deviates from the TWR evaporation line, indicate that there are other sources of groundwater recharge in the sand layer (the influence of preferential flow and lateral flow). Finally, we modified the sentence to "*There were few **samples** in the **sand layers** where C1 showed extreme deviation from the TWR evaporation line (labeled in Fig. 6a).* " and replace "aquifer" with "sand" in Fig. 6.

[Figure]

***Fig. R5    Modified Figure 6.***

**RC1:** In general, the entire section 3.3 is hard to read and difficult to understand. I suggest trying to re-write this part more clearly. In its present form this section looks to suggest that there is no influence of sediments' heterogeneity and circulation patterns on stable isotopes changes because all the pore water derives from zenithal infiltration. Is this Your idea? If so You should discuss the reasons for that.

**Reply:** We'll consider rewriting that part.

**RC1:** Section 3.4. In this section You use data from "shallow groundwater" (80- 120 m deep). Where did You collect samples for these analyses? You plot data from "shallow" and "deep" groundwater in Fig.8. Where do they come from? Your study is limited to the uppermost 30 m, which is the relationship with the lower hydrostratigraphic section and groundwater? This part is totally obscure to the reader.

**Reply:** The collection of shallow and deep groundwater samples is mentioned in section 2.2 (Italicize below). We sampled wastewater at the study location in June 2017, and sampled both shallow groundwater (80-120m) and deep groundwater (120m-300m) over a 2.5km area along the Tang River. Shallow groundwater is the main source of irrigation water, the main recharge of the upper shallow groundwater. And deep groundwater is only used to compare the difference with shallow groundwater. So the text refers more to the upper shallow groundwater and shallow groundwater.

……*We sampled wastewater at the study location in June 2017, and sampled both shallow groundwater (80-120m) and deep groundwater (120m-300m) over a 2.5km area along the Tang River.*……

**RC1:** Line 316: "Similarity"? Did You mean "similarly"?
**Reply:** Yes, it should be " similarly ".

**RC1:** Line 321: "… relationships for…" did You mean relationships between?
**Reply:** Yes, it should be " relationships between ".

**RC1:** Caption to Fig.7: may be the curve in C1 was not shown? Otherwise change C0 to C1 in legend.
**Reply:** Yes, it should be " C1 ".

**RC1:** Lines 327-330: how can You exclude remote provenance for groundwater? Did You ever collect data from the lake waters?
**Reply:** Although the groundwater flows from east to west at present, according to the hydrogeological conditions of the study area (the aquifer permeability coefficient is 4.66m/d, the effective porosity is 0.26, the hydraulic gradient is 0.5‰-1.3‰) we calculated the average recharge time from the Baiyangdian Lake to the study area with a distance of 9.5km from Baiyangdian Lake (the actual flow length >9.5km) by using the Darcy's law. It was estimate that it takes 1117-2904 years for the lake water recharging to the study area. Moreover, during the sewage storage period of the TWR (1977-2015), the wastewater of the TWR leached and recharged to local upper shallow groundwater as well as precipitation, irrigation water, and lateral groundwater which is mainly from the large farmland north of TWR. As a result, the groundwater of the study area has not been affected by the lake water. We have mentioned the change of groundwater flow in section 3.1, and we also added the aquifer permeability coefficient, effective porosity, study location and recharge source discussion in the revised manuscript in order to express it more clearly. Details are shown in the italicized statements below.

*The monthly variations in water table levels for all monitoring wells were calculated according to the elevation of each well and monitored USGW levels (Fig. S3). At the beginning of the monitoring period, groundwater levels along the transect decreased from the north sites to the south sites*

*(C6>C5>C3>C4>C2>C1>C0), indicating a groundwater flow from north to south. Subsequently, due to the removal of the TWR effluent and the recharge of Baiyangdian Lake, the groundwater level at the study location continues to drop and the difference in water level between monitoring points on the cross section is small, and the groundwater flows from east to west during the late monitoring period ( Fig. 1c). Although the USGW flows from east to west at present, the sand layer permeability coefficient is 4.66m/d, the effective porosity is 0.26, the hydraulic gradient is 0.5‰-1.3‰, and the study location is 9.5km from lakeside (the actual flow length >9.5km), Baiyangdian Lake water cannot feed the groundwater at the study location during the monitoring period, according to Darcy's law.*

**RC1:** Caption to Fig.8: "…with depth of 100-120 m." and "…with depth ranging from 120 m – 300 m". At first, I thought it was a typewriting mistake (m instead of cm), then I had to get back to the former sections to understand that You are here introducing data from elsewhere, tens to hundreds of m below Your cross-section, but You did not mention these samples before (for instance in the methods section). This is misleading. You also mix the terms shallow, deep, soil, groundwater, aquifer giving them different meanings in different sentences. The only clear thing to the reader is that You are not considering the true shallow and deep hydrostratigraphy of the site, that instead should be the basis to interpret all of Your isotope and hydrochemical data, to give sense to the hysteresis loops, and to understand the mixing and recharge processes through the heterogeneous and anisotropic porous system.

**Reply:** The shallow groundwater (80-120) was mainly used for agricultural irrigation, which is one of the the main recharge sources for the upper shallow groundwater (<30m). While the deep groundwater (120-300m) was mainly used as drinking water which is mainly used to compare the difference of isotopes between shallow groundwater and deep groundwater. To facilitate the distinction, we have added a note at the end of 2.1. Additionally, we have added a hydrogeological section in supplementary file (Fig.S1). And about the aquifer and groundwater may not be readable within some sentences, we have made some adjustments. For example, in the description of porewater isotopes and migration pathways, we use 'sand layers' instead of 'aquifers', while we continue to use the term 'aquifer' in the description of regional hydrogeological conditions.

**RC1:** Lines 345-347: what do You mean by "upper stream of the transect"? Do You mean the northernmost sampling site? You stated that the phreatic surface gradient is to the WSW (see also Fig.1). On the contrary, at lines 223-226 You suggest southwards groundwater flow followed by a N-wards (?) reversal during late sampling. Based on which gradient of which groundwater body within which aquifer do You infer lateral recharge in this sentence? Definitely a more clear and complete characterization of the site is necessary. I suggest again to complete the dedicated paragraph, that in its present form contains less than two lines describing aquifer stratigraphy, to present all the relevant features (hydrology and geology) in order to avoid contradictions and misleading interpretations.

**Reply:** Thank you for your suggestion. We have given a detailed description of the changes of groundwater flow. Here it should be explained in the heterogeneity of the vadose zone. Compared with C0, the thickness of the silty clay layer is smaller in the vadose zone at C6, causing a larger proportion of irrigation water to infiltrate into the groundwater.

**RC1:** Lines 346-347: are You talking of shallow or deep groundwater? (this question holds also for all the former part of this section). I would expect a progressively minor contribution from irrigation downwards, with a consequent increase of the precipitation signal, that might simply indicate that the lower aquifer (within Your cross-section) receives a remote recharge, eventually also from the lake.

**Reply:** All discussions of groundwater in the manuscript are about the upper shallow groundwater, which we will note at the end of section 2.1 to clarify this to the reader. Also as elaborated in the previous reply, there is no remote recharge at this location during the monitoring period.

**RC1:** Section 4. In Your discussion You never comment on any eventual changes with depth through the 5 sites.

**Reply:** Although in this paper we rarely discuss the variation in depth, it is important to understand the actual recharge process, which will be the focus on our next work. However, in this manuscript we focused more on the effect of matrix flow and preferential flow co-recharge on groundwater quality. The distribution of pore water hydrogen and oxygen isotopes in the five sediment profiles was used to identify differences in upper shallow groundwater recharge processes, and recharge rates were obtained based on variations in profile depth when combing the historical hydrologic conditions at the surface.

**RC1:** Section 4.1: revise the sentence at lines 391-392.

**Reply:** The depth of information in the previous sentence was confusing. Here is the revised statement.

*Profiles C1 and C5 were located outside of the TWR embankment. A T test showed that isotopes in the C1 and C5 profiles were similar to that of C3. This indicated that the local lateral flow of TWR sewage through the vadose zone affects the stable isotopes of porewater in C5 and C1. This was verified by a field survey that found the polluted sludge distributed along the silt clay layer under 450 and 875 cm for C5 and C1, respectively, which also supported the lateral flow of sewage in unsaturated zone (Fig. 3a).*

**RC1:** Section 4.2. Line 439-441: sorry, I am lost. Until here I thought (Fig.1 and Fig.3 and main text) that C3, C2 and C4 (the last two never commented) were within the embankments, C1 and C5 on the embankments and C0 and C6 were outside the embankments. No reference to the natural Tanghe River channel width has been ever given until here and no evidence of the presence of shallow permeable sediments in the unsaturated zone relating to the river channel has been provided (Fig.1 and 3, main text). I already suggested to better describe the setting of the study site, this point of the discussion strongly highlights this need.

**Reply:** We have added more information about the setting of the study area in section 2.1 of the revised manuscript. C1 and C5 are located on the outer side of the TWR embankment, adjacent to the embankments. In addition, in section 2.2 we mentioned that the collection of sediment samples and analysis of stable isotopes of porewater were only conducted for C0/C1/C3/C5/C6, and the collection of sediment samples for C2 and C4 were only used for observing sediment structure. That's why C2 and C4 are not mentioned and analyzed in later sections.

[Figure]

(a) Sludge in sediments  (b) Silty clay  (c) Silt  (d) Silty clay  (e) Sand  (f) Silty clay

**RC1:** Lines 440-441. This sentence is totally obscure to me: how do "groundwater affect the aquifers … by lateral flow"? It really looks that the missing scheme of groundwater circulation through the heterogenous and anisotropic hydrostratigraphy of the study site affects this discussion.

**Reply:** The sentence was rephrased as *"Eventually the groundwater downstream is affected by the rapid lateral flow of the sand layer."* It means that a highly permeable sand layer can affect the water quality of downstream groundwater through lateral flow.

**RC1:** Fig.12: please report the depth of the model making it comparable to Fig.3. In this picture You draw down to 80-120 m, so which part of this picture overlaps the sampled section (30 m thick, no more than 20 m below sea-level). In this picture You show 4 divisions in the subsurface, with different colors. Do they correspond to some layers in Fig. 3? Is there some relation between the (questionable) hydrostratigraphy shown in Fig.3 and this picture? What do You mean by Upper shallow groundwater? Does this term relate to the terms You used in the former main text? Where do You derive information about the existence of a shallow groundwater at 80-120 m? (and why is this "shallow" at this depth?). Why lateral flow is considered only for the lowermost aquifer (of Fig.3) and not also for the uppermost one? Any data about lake water, that is one source of lateral recharge and sustains groundwater circulation of the shallow aquifers?

**Reply:** This is a conceptual diagram formed by summarizing and generalizing the results and discussion sections. It shows the recharge process of the preferential flow and matrix flow of the upper shallow groundwater. In the figure we have labeled the upper shallow groundwater depth (<30m) and shallow groundwater (80-120m), however, the shallow groundwater is used as a source of irrigation water recharging to upper shallow groundwater. Moreover, this concept diagram is a simplification of Figure 3. In addition, the lake is not a recharge source during the monitoring period at the study area, as we have discussed in the previous reply.

**RC1:** Line 474. I would not say that You described the pathways of pollutants, since the flow paths through the aquifer architecture are not described because the study is 2D.

**Reply:** Agree. In this sentence, we are expressing the processes from the surface to the upper shallow groundwater. We will revise the manuscript later to make the purpose of our study more prominent.

**RC1:** Lines 476- 477: since the introduction, the features of this "alluvium river aquifer" were not described. It looks to be mostly made of very fine-grained sediments, typical of an anastomosed river mouth plausibly under tidal influence that should be characterized by very low K values, but no estimates of the K distribution have been provided through the (questionable and poorly supported) hydrostratigraphic cross-section. Occasionally, the alluvium within the Tanghe River channel (that is neither mapped nor described) is considered as a more permeable unit than the adjacent ones. Definitely the sediments and the resulting hydrostratigraphy should be clarified to locate the isotope and hydrochemical data in their correct space-time setting to support the discussion and conclusions You draw. Please, also watch the use of the term "alluvium river" that does not exist as far as I know.

**Reply:** We described the characteristics of the "alluvium aquifer" in the first paragraph of the introduction, including its rapid renewal rate, vulnerability to contamination, and complex recharge processes, as shown in the italicized text. *Groundwater is the primary water supply in many regions that have rapidly expanding water requirements including urban, industrial, and agricultural production. Groundwater is particularly important in alluvium aquifers, where groundwater renewal rates are generally high (Sun et al., 2021; Ma et al., 2019) and thus, functioning as potable water sources in arid, semi-arid, and semi-humid areas. However, these aquifers are also susceptible to depletion and contamination, with recharge rate and dominant flow processes determining their level of vulnerability (De Vries and Simmers, 2002). Groundwater recharge is influenced by complex ecohydrology processes that are controlled by geology, meteorology, morphology, and vegetation. Identifying recharge sources and mechanisms is important to develop strategies to prevent groundwater pollution (Jasechko, 2019).*

And we will emphasize again in section 2.1 the high permeability and inhomogeneity of "alluvium aquifer" in the study area, as shown in Fig. R8. *According to the regional hydrogeological survey, the 30 m depth range consists of Quaternary Holocene and Late Pleistocene alluvial sediments with a high level of permeability and heterogeneity, in which the unsaturated zone interlayered with silt and silty clay, and the saturated zone comprises silt and sand with local inclusions of thick layers of silty clay.*

Also, thanks for the reminder about the improper application of "alluvium river aquifer". "alluvium aquifer" should be a more technical term, so we will replace "river alluvium aquifer" and "alluvium river aquifer" in the manuscript with "alluvium aquifer".

**RC1:** Lines 480-481: You did never discuss to which parts of the hydrostratigraphic section do these values refer.

**Reply:** This value refers to our calculated infiltration rate of C0, C3 and C6, calculated via the history of the sewage reservoir and agricultural irrigation and the pore water isotopic characteristics of the sediment profiles. A detailed discussion can be seen in section 4.1 (Italicize below), where we present a careful statement and provide a comparison with the results of other studies on the region.

*As Fig. 5, and 6 show, the stable isotopes in the five sediment profiles indicated differences in infiltration processes and water sources in porewater when sewage remained in the TWR channel. All profiles have an obvious isotopic peak point for indicating the vertical variation in sources. These points were 900, 450, 650, 875 and 750 cm for C6, C5, C3, C1, and C0, respectively (Fig. 5),*

*suggesting infiltration and mixing of different sources. For example, the C3 profile contained sewage from the TWR and the enrichment of δ2H and δ18O from deep to shallow depths indicates infiltration and mixing of the more enriched sewage in the TWR after 2011 with previously recharged sewage (Fig. 3). The enrichment of stable isotopes in deep layers (>650 cm depth) and the quickly enriched isotopes in shallow layer (<650 cm depth) of the C3 profile were in accordance with the stable isotopes in the sewage of the TWR before 2011 and the greater enrichment from 2011 to 2017, respectively. According to the recharge depths (650 cm) and the recharge time of 6 years from 2011 to 2017, the average recharge rate of sewage was estimated to be 109 cm/year for C3. Similarly, the vertical distributions of isotopes in the farmlands with the C0 and C6 profiles revealed the enriched sewage that was irrigation before 2005 and SGW irrigation with depleted isotopes (mean -62.5 and -7.9‰ for δ2H and δ18O, respectively) from 2005 to 2017. According to the recharge depths of 750 and 900 cm and recharge time of 12 years for C0 and C6, respectively, we estimated recharge rates of 63 and 75 cm/year for C0 and C6, respectively. These recharge rates are in the range of 41 to 200 cm/year as estimated by previous research completed in the alluvium plain of the North China Plain (Min et al., 2018; Wang et al., 2019). Due to the high sediment moisture under sewage (Fig. 5), the recharge rate of C3 was higher than the cropped fields of C0 and C6. The relatively greater recharge rate in the cotton field of C6 compared to the wheat/maize field of C0 can be attributed to the relatively thinner silt layer in C6.*

[Figure]

**Figure 5.** Vertical distribution of stable isotopes and gravimetric water content (Wg) in porewater for profiles C0-C6.